# You Get What You Give: Reciprocally Fair Federated Learning

**Aniket Murhekar** [1]   **Jiaxin Song** [2]   **Parnian Shahkar** [3]   **Bhaskar Ray Chaudhury** [1 2]   **Ruta Mehta** [1]

## Abstract

Federated learning (FL) is a popular collaborative learning paradigm, whereby agents with individual datasets can jointly train an ML model. While higher data sharing improves model accuracy and leads to higher payoffs, it also raises costs associated with data acquisition or loss of privacy, causing agents to be strategic about their data contribution. This leads to undesirable behavior at a Nash equilibrium (NE) such as *free-riding*, resulting in sub-optimal fairness, data sharing, and welfare. To address this, we design $\mathcal{M}^{Shap}$, a budget-balanced payment mechanism for FL, that admits Nash equilibria under mild conditions, and achieves *reciprocal fairness*: where each agent's payoff equals her contribution to the collaboration, as measured by the *Shapley share*. In addition to fairness, we show that the NE under $\mathcal{M}^{Shap}$ has desirable guarantees in terms of accuracy, welfare, and total data collected. We validate our theoretical results through experiments, demonstrating that $\mathcal{M}^{Shap}$ outperforms baselines in terms of fairness and efficiency.

## 1. Introduction

Federated learning (FL) provides an effective distributed learning paradigm where a group of agents holding local data samples can train a joint machine learning model. The paradigm has extensive applications, including autonomous vehicles (Elbir et al., 2020) and digital healthcare (Dayan et al., 2021; Xu et al., 2021a).

The success of federated learning hinges on the availability of diverse, high-quality data from a variety of agents for effective training. However, data sharing is often costly due to factors such as acquisition costs (Tu et al., 2022), computational expenses, and privacy concerns (Chen et al., 2020). As a result, agents may act strategically and reduce their data contributions, particularly if they bear high data sharing costs. This can lead to undesirable outcomes in terms of both *fairness* – where agents receive benefits disproportionate to their data contributions – and *efficiency* – resulting in low data sharing, reduced total welfare, and suboptimal learning outcomes. To address these challenges, it is crucial to design mechanisms for federated learning that incentivize participation from strategic agents and also ensure fairness and efficiency at stable solutions (like a Nash equilibrium).

To this end, (Karimireddy et al., 2022) introduce the *data sharing incentivization framework* in FL, where each federating agent's net utility can be measured as the difference between the accuracy gained in the federation and the cost of data-sharing, and the agents are strategic about data contributions. (Karimireddy et al., 2022) then consider a mechanism based on contract theory, where each agent receives a personalized model whose accuracy is tuned based on data contribution of the agent to the training (*data-share maximizing* mechanism). In a similar spirit, (Murhekar et al., 2023) study a mechanism in the same framework with payments, where agents may be charged/rewarded with money, such that any Nash equilibrium (NE) achieves the maximum utilitarian welfare possible under the mechanism (*welfare maximizing* mechanism). Observe that both the foregoing guarantees at NE are *efficiency* guarantees: they ensure maximal data gain or maximal welfare gain out of the federation.

In this paper, we investigate mechanisms for FL, which are *fair* in addition to being efficient. Our notion of fairness is *reciprocity*: a mechanism is considered reciprocally fair if each agent is guaranteed a "reciprocal" final utility commensurate with her contribution to the learning process. Naturally, a reciprocal mechanism incentivizes participation from agents holding valuable data, which is in line with the goals of a mechanism designer for FL. Empirical evidence from behavioral economics (Fehr & Schmidt, 2006) further shows that in contrast to the *self-interest hypothesis*[1], users seem to trust reciprocal mechanisms, especially in bargaining and co-operative environments (like FL), and this trust

---
[1]Department of Computer Science, University of Illinois, Urbana-Champaign, Urbana, USA [2]Department of Industrial Systems Engineering, University of Illinois, Urbana-Champaign, Urbana, USA [3]Department of Computer Science, University of California, Irvine, Irvine, USA. Correspondence to: Bhaskar Ray Chaudhury <braycha@illinois.edu>, Ruta Mehta <rutamehta@cs.illinois.edu>.

*Proceedings of the 42nd International Conference on Machine Learning*, Vancouver, Canada. PMLR 267, 2025. Copyright 2025 by the author(s).

---
[1]Only material self-interest motivates all user participation

can lead to large voluntary participation.

To formalize the foregoing statements in the context of FL, we define the *reciprocity* of a mechanism as the minimum over all Nash equilibria, the minimum over all agents, of the ratio of the benefit that an agent receives *from* the mechanism,[2] and her contribution *to* the benefit (accuracy) of all agents. In our work, we measure an agent's contribution to the total accuracy of all agents using the classical notion of *Shapley value* (Shapley, 1953) from cooperative game theory, as done in (Bhaskara et al., 2024; Wang et al., 2019; Sim et al., 2020; Ghorbani & Zou, 2019; Agarwal et al., 2019). We note that reciprocity $r$ of any budget-balanced mechanism lies in $[0, 1]$, with higher $r$ implying better fairness. We observe that the efficiency-focused mechanisms in (Karimireddy et al., 2022) and (Murhekar et al., 2023) are not reciprocal (see Example 7 and Section 5), which brings us to the driving question of the paper:

*Are there reciprocal mechanisms that admit a Nash equilibrium? Do the Nash equilibria also provide efficiency guarantees in terms of total data contributed (as in (Karimireddy et al., 2022)), and total welfare achieved (as in (Murhekar et al., 2023))?*

**Our Contributions.** We propose a budget-balanced mechanism for federated learning called $\mathcal{M}^{\mathsf{Shap}}$, which is *reciprocally fair* and admits *efficient* Nash equilibria.

- $\mathcal{M}^{\mathsf{Shap}}$ admits Nash equilibria, which can be computed efficiently through *stochastic best response dynamics*, under mild assumptions on agents accuracy and cost functions.

- $\mathcal{M}^{\mathsf{Shap}}$ is *cost-agnostic*, meaning it does not require knowledge of each agent's private data-sharing cost, thus alleviating the burden of cost-verification for the central server. Importantly, unlike previous cost-based payment schemes like (Murhekar et al., 2023) that unfairly reward agents with high-acquisition-cost and low-quality data, $\mathcal{M}^{\mathsf{Shap}}$ only rewards agents with high-quality data.

- $\mathcal{M}^{\mathsf{Shap}}$ is *fair*: it is fully reciprocal, satisfies equal treatment of equals, and is individually rational.

- Surprisingly, $\mathcal{M}^{\mathsf{Shap}}$ admits *efficient Nash equilibria* (NE), despite being designed for fairness. In particular, there is *no other mechanism* that simultaneously Pareto-dominates the final data share and total welfare of $\mathcal{M}^{\mathsf{Shap}}$. In other words, for every data share $s$ that Pareto-dominates the data share at an NE of $\mathcal{M}^{\mathsf{Shap}}$, the total welfare at $s$ will be strictly lower. Conversely, for every data share $s$ where the total welfare is greater than that achieved at a NE of $\mathcal{M}^{\mathsf{Shap}}$, there exists at least one agent whose data share is strictly lower in $s$. We then define

---

[2]Since this is the benefit the mechanism provides, this excludes the cost which is private to an agent

metrics to measure the efficiency of a mechanism, namely, *data gain* and *accuracy gain* (formally defined in Sec. 3). For structured payoff and cost functions used in the literature (Karimireddy et al., 2022; Murhekar et al., 2023), we establish strong lower bounds ($\Omega(\sqrt{n})$) on the data gain and accuracy gain of $\mathcal{M}^{\mathsf{Shap}}$, which improve as the number of agents $n$ increases.

- We empirically evaluate our mechanisms on five datasets: MNIST, FashionMNIST, CIFAR-10, Lumpy Skin Disease, and a synthetic quadratic regression dataset. We compare the performance of the Nash equilibria of $\mathcal{M}^{\mathsf{Shap}}$ and two baseline mechanisms – the mechanism $\mathcal{M}^0$ without payments, and the welfare-maximizing mechanism (Murhekar et al., 2023). Our mechanism outperforms in metrics of reciprocity, data gain, accuracy gain, and total welfare (see Table 1).

- We design a distributed FL protocol `FedBR-Shap` to approximately compute the NE of $\mathcal{M}^{\mathsf{Shap}}$. Unlike prior work (Karimireddy et al., 2022; Murhekar et al., 2023), our protocol relies exclusively on gradient information, eliminating the need for sharing actual data points and uses actual model accuracies instead of assuming a closed form of accuracies, marking a departure from prior work.

Although our theoretical results apply broadly, our proposed method is especially well suited to cross-silo federated learning (see (Zeng et al., 2022)), where only a moderate number of clients participate in collaboration, and thereby Shapley-value computations remain tractable.

## 1.1. Related Work

The subject of incentives in federated learning has received substantial attention (see (Tu et al., 2022) for a detailed survey) as the federating agents indeed have benefits and costs (communication and computation (Tu et al., 2022), privacy loss due to generative adversarial attacks (Chen et al., 2020)). Several concepts from game theory (Stackelberg games (Khan et al., 2020; Pandey et al., 2019), non co-operative games (Zou et al., 2019; Cheng et al., 2021), auctions (Roy et al., 2021)) have been adopted for incentivizing participation in FL. We remark that rewarding agents according to their contribution levels has been well motivated and studied in FL (Wang et al., 2019; Sim et al., 2020; Zhang et al., 2020; Yu et al., 2020). However, the crucial difference is that our focus is to design a mechanism that incentivizes *strategic agents*, i.e., agents who strategize their data contributions based on the rewards they get from the federation so that desirable fairness and welfare guarantees are achieved at NE (in line with the works of (Karimireddy et al., 2022; Murhekar et al., 2023) studying NE properties, and (Chen et al., 2023; Cai et al., 2014) incentivizing the supply of high-quality data). In contrast, Chaudhury et

al. (Chaudhury et al., 2022) assume non-strategic agents who share their complete datasets, and under this setting guarantee coalition stability.

The Shapley value (Shapley, 1953) is a well-studied concept from cooperative game theory, used to distribute the benefit of cooperation among participating agents. In federated learning, Shapley value has been used to measure the contribution of participants (Wang et al., 2019; Xu et al., 2021b), interpret models (Wang, 2019), value data (Wang et al., 2020), and allocate profit (Song et al., 2019).

Finally, to illustrate practical payment mechanisms in FL, we highlight blockchain-based approaches that reward participants according to their individual contributions. FedCoin (Liu et al., 2020) uses a proof-of-Shapley protocol, while FedToken (Pandey et al., 2022) distributes tokens based on performance. Both require an initial budget, unlike our budget-balanced approach, which penalizes poor data quality and rewards high-quality data. Also in IoT, BOppCL (Li et al., 2024) incentivizes vehicles in intelligent transportation systems, rewarding those with more useful data via cryptocurrency.

## 2. Preliminaries

**Problem formulation.** We study the data-sharing incentivization framework introduced by (Karimireddy et al., 2022). There is a set $N = [n]$ of $n$ agents who wish to solve a learning problem by training a joint model, and a central server that coordinates among them. Agent $i$ transmits a set $T_i \sim \mathcal{D}_i^{s_i}$ of $s_i$ i.i.d. data points sampled from the distribution $\mathcal{D}_i$ of data samples available to $i$. Let $S_i = [0, \tau_i]$ denote the set of feasible number of samples agent $i$ can contribute, and let $\mathcal{S} := \bigtimes_j S_j$. For a *sample vector* $\boldsymbol{s} = (s_1, s_2, \ldots, s_n) \in \mathcal{S}$ specifying the data contributions of the agents, the central server returns a model trained using the data set $\bigcup_i T_i$, which has size $\|\boldsymbol{s}\|_1 = \sum_i s_i$.

Every agent receives a *payoff* or benefit from federation as well as incurs a *cost* of sharing data samples. We measure the payoff to agent $i$ from the jointly trained model using a payoff function $a_i : \mathcal{S} \to \mathbb{R}_{\geq 0}$. Generally, we assume the payoff $a_i(\boldsymbol{s})$ represents the *accuracy* of the jointly trained model at the contribution level $\boldsymbol{s}$ on agent $i$'s learning task. However, our model allows for more general payoff functions. Moreover, each agent $i$ incurs a *cost* associated with data sharing modeled using a cost function $c_i : S_i \to \mathbb{R}_{\geq 0}$. Thus, each agent obtains a *utility* $u_i(\boldsymbol{s})$ at sample vector $\boldsymbol{s}$ given by the difference between the payoff received and the cost incurred, i.e.,

$$u_i(\boldsymbol{s}) = a_i(\boldsymbol{s}) - c_i(s_i). \tag{1}$$

We now discuss some canonical examples and properties of the payoff and cost functions.

### 2.1. Payoff and cost functions: examples and properties

**Payoff functions.** We assume that each payoff function $a_i$ is bounded, non-decreasing and *concave* in the contribution of all agents. This is a standard assumption in literature (Blum et al., 2021; Karimireddy et al., 2022; Murhekar et al., 2023), and captures decreasing marginal returns on increased data sharing. Below we present canonical examples of payoff functions in common learning frameworks that satisfy the above assumptions.

*Example* 1 (Generalization bounds from general PAC learning (Mohri et al., 2018)). Consider a general learning problem of learning a model $h$ from a hypothesis class $\mathcal{H}$ which minimizes the expected error $R(h) = \mathbb{E}_{(x,y)\in\mathcal{D}} e(h(x), y)$ over a data distribution $\mathcal{D}$, for some loss function $e(\cdot)$. Given $m$ i.i.d. samples from $\mathcal{D}$, the empirical risk minimizer (ERM) is the model $h_m = \arg\min_{h\in\mathcal{H}} \sum_{\ell\in[m]} e(h(x_\ell), y_\ell)$. (Mohri et al., 2018) prove that with high probability, the error of $h_m$ is bounded:

$$1 - R(h_m) \geq a(m) := a_0 - \frac{4 + \sqrt{2k(2 + \log(m/k))}}{\sqrt{m}}, \tag{2}$$

where $(1 - a_0)$ is the accuracy of the optimal model from $\mathcal{H}$, and $k$ is a (constant) measure of the difficulty of the learning problem depending on $e(\cdot)$ and $\mathcal{H}$.

In fact, there are other examples like *linear, random discovery, and random coverage* based accuracy functions (Blum et al., 2021) that also satisfy our desired criterion, and they are discussed in more detail in Appendix B.

*Example* 2 (Empirical evidence). (Kaplan et al., 2020) and (Henighan et al., 2020) discuss empirical scaling laws relating to cross-entropy loss on neural and large-language models for a number of ML tasks relating to language, image, and video. They observe that the loss scales with the dataset size $m$ as a power law $\ell(m) = \alpha \cdot m^{-\beta}$, for some parameters $\alpha, \beta \in (0, 1]$.

The above examples indicate the dependence of the accuracy $a(m)$ on a learning task given $m$ data points varies as $1 - \alpha \cdot m^{-\beta}$, both in theory and in practice.

**Cost functions.** We assume that each cost function $c_i$ is continuous, non-decreasing, and convex in $s_i$. This is also a standard assumption (Blum et al., 2021; Karimireddy et al., 2022; Murhekar et al., 2023), and captures non-decreasing marginal costs (Li & Raghunathan, 2014).

### 2.2. Nash Equilibrium

Recall from Equation (1) that the net utility of agent $i$ is the difference between the payoff received and the cost incurred, *i.e.*, $u_i(\boldsymbol{s}) = a_i(\boldsymbol{s}) - c_i(s_i)$. Since agents are net-utility maximizers, the goal of an agent is to strategically decide how many samples to contribute so that her net utility is max-

imized. To analyze agent strategic behavior arising in FL, we use the standard game-theoretic concept of Nash equilibrium ((Nash, 1951)). Intuitively, the NE is a stable state of the system where no agent can increase their utility by unilaterally changing their data contribution level. Formally:

**Definition 2.1** (Nash equilibrium (NE)). A sample vector $\boldsymbol{s} \in \mathcal{S}$ is a Nash equilibrium if for every $i \in N$, and every $s_i' \in S_i$, we have $u_i(\boldsymbol{s}) \geq u_i(s_i', \boldsymbol{s}_{-i})$ where $(s_i', \boldsymbol{s}_{-i}) = (s_1, \ldots, s_i', \ldots, s_n)$.

The concept of *best response* provides an alternate, dynamic view of Nash equilibrium. Facing the sample vector $\boldsymbol{s}_{-i}$ of agents other than $i$, the set $f_i(\boldsymbol{s}_{-i})$ of contribution levels of agent $i$ that maximizes her utility is defined as the best response set of agent $i$:

$$f_i(\boldsymbol{s}_{-i}) = \arg\max_{x \in S_i} \left\{ a_i(x, \boldsymbol{s}_{-i}) - c_i(x) \right\} \subseteq S_i.$$

The *best response correspondence* $f$ is the set-valued function given by $f : \mathcal{S} \to \bigtimes_i 2^{S_i}$, where $[f(\boldsymbol{s})]_i = f_i(\boldsymbol{s}_{-i})$. Thus, Nash equilibria are fixed points of this correspondence.

**Proposition 2.2.** *A sample vector $\boldsymbol{s} \in \mathcal{S}$ is a Nash equilibrium if and only if it is a fixed point of the best response correspondence, i.e., $\boldsymbol{s} \in f(\boldsymbol{s})$.*

(Murhekar et al., 2023) proved that an FL problem admits a NE with concave payoffs and convex costs, but not without these assumptions.

### 2.3. Shapley value

The Shapley value (Shapley, 1953) is a classic solution concept from cooperative game theory that specifies a *fair* way of distributing the surplus generated in a cooperative game. We adapt this concept in the federated learning context as follows. The total surplus generated at a sample vector $\boldsymbol{s}$ is the cumulative payoff to all agents given by $A(\boldsymbol{s}) := \sum_{i=1}^{n} a_i(\boldsymbol{s})$. Let $\boldsymbol{s}[X]$ denote the sample vector restricted to agents in $X$, i.e., $\boldsymbol{s}[X]$ is the vector $s'$ given by $s_i' = s_i$ for $i \in X$ and $s_i' = 0$ for $i \notin X$. To measure the "contribution" of agent $i$ to $A(\boldsymbol{s})$, we compute for every coalition $X \subseteq N \setminus \{i\}$ of agents the marginal gain $A(\boldsymbol{s}[X \cup \{i\}]) - A(\boldsymbol{s}[X])$ of adding $i$ to $X$, and average it over all ways of forming the coalition. This represents the *Shapley value of federation* $\varphi_i^A(\boldsymbol{s})$ at $\boldsymbol{s}$, and is formally expressed as:

$$\varphi_i^A(\boldsymbol{s}) = \frac{1}{n} \cdot \sum_{X \subseteq N \setminus \{i\}} \binom{n-1}{|X|}^{-1} \cdot \left( A(\boldsymbol{s}[X \cup \{i\}]) - A(\boldsymbol{s}[X]) \right)$$

(3)

The following lemma (adapted from the well-known properties of Shapley values) shows that the Shapley values

distribute the cumulative payoff (see Appendix C).

**Lemma 2.3.** *For any $\boldsymbol{s} \in \mathcal{S}$, $A(\boldsymbol{s}) = \sum_{i \in N} \varphi_i^A(\boldsymbol{s})$.*

## 3. Mechanisms and Metrics for FL

Since NE is an intuitive and well-established solution concept that is guaranteed to exist, it is natural to examine the properties of a NE in the context of federated learning.

*Example* 3. Consider two agents with identical payoff function $a(\boldsymbol{s}) = 1 - (\|\boldsymbol{s}\|_1 + 1)^{-1}$ and linear cost functions given by $c_1(s_1) = 0.1s_1$ and $c_2(s_2) = 0.25s_2$. The only NE is given by $\boldsymbol{s}^0 = (2.16, 0)$.

The above example shows that NE can be quite far from desirable: at NE, only agent 1 contributes data samples while agent 2 does not contribute at all and *free-rides*. Similar examples illustrate that NE can be far from optimal in terms of fairness, overall data shared (Blum et al., 2021), and overall agent welfare (Murhekar et al., 2023), all of which are part of the desiderata for federated learning solutions. To address some of these issues, prior works (e.g. (Blum et al., 2021; Murhekar et al., 2023)) studied mechanisms for FL that incentivize agents to contribute more data or admit NE with good welfare. Like (Murhekar et al., 2023), our work focuses on *mechanisms with payments*, formalized as follows.

**Definition 3.1** (Mechanisms with payments). A mechanism $\mathcal{M}$ models the interaction between the central server and the participating agents. First, the server publishes the mechanism by specifying a *payment scheme* $p$ which indicates the payment $p_i(\boldsymbol{s})$ to each agent $i$ at the sample vector $\boldsymbol{s}$. Second, the agents observe the mechanism, decide their data contribution levels $\{s_i\}_{i \in N}$ individually, and transmit the data to the server. Third, the server computes and returns a model trained on data contributed by all agents at the sample vector $\boldsymbol{s}$, and returns the payment $p_i(\boldsymbol{s})$ to each agent $i$[3].

Thus, under a mechanism $\mathcal{M}$ with payment scheme $p$, the resulting utility of an agent $i \in N$ is given by $u_i(\boldsymbol{s}) = a_i(\boldsymbol{s}) - c_i(s_i) + p_i(\boldsymbol{s})$.

A mechanism is said to weakly balance its budget if the total payment to the agents is not positive. A stronger condition is budget-balance, which requires that the total payment to the agents is zero, i.e., the central server operates on no profit or loss.

**Definition 3.2** (Budget-balanced mechanism). A mechanism is said to be weakly budget-balanced if for every $\boldsymbol{s} \in \mathcal{S}$, $\sum_{i \in N} p_i(\boldsymbol{s}) \leq 0$. If the latter is an equality, then the mechanism is budget-balanced.

---

[3]Note that if $p_i(s) < 0$, agent $i$ must pay the server. We can enforce this by transmitting the model only *after* receiving payment from the agent. Thus for simplicity, we assume agent $i$ can only strategize on $s_i$.

We now define some fairness and efficiency metrics to measure the quality of a mechanism $\mathcal{M}$ and its set of NE denoted by $\mathsf{NE}(\mathcal{M})$. Naturally, we are only interested in mechanisms that admit a NE.

### 3.1. Metrics to Evaluate FL Mechanisms

**Individual rationality.** A mechanism $\mathcal{M}$ is said to be individually rational if every agent gets non-negative utility at its NE, i.e., for all $s \in \mathsf{NE}(\mathcal{M})$, for all $i$, $u_i(s) \geq 0$. Thus, no agent can receive a negative utility by participating in an individually rational mechanism.

**Fairness metrics.** By participating in the mechanism, each agent contributes towards the benefit of other agents while also reaping benefits from others. At a sample vector $s$, the contribution of an agent $i$ from the mechanism is $a_i(s) + p_i(s)$ while the contribution to the mechanism is $\varphi_i^A(s)$. The following fairness metric, termed Reciprocity, measures the degree to which the worst NE of a mechanism *reciprocates* the contribution of any agent.

**Definition 3.3** (Reciprocity of a mechanism)**.** The reciprocity of a mechanism $\mathcal{M}$ is defined as:

$$\mathsf{Reciprocity}(\mathcal{M}) = \min_{s \in \mathsf{NE}(\mathcal{M})} \min_{i \in N} \frac{a_i(s) + p_i(s)}{\varphi_i^A(s)}. \quad (4)$$

Thus, a mechanism $\mathcal{M}$ with reciprocity $r < 1$ has some NE $s \in \mathsf{NE}(M)$ that reciprocates some agent $i$ less than her contribution at $s$, i.e., is unfair. On the other hand, $r > 1$ implies that every NE reciprocates every agent strictly more than they contribute. The following lemma proves the intuitive fact that such a mechanism must make a total positive payment to the agents.

**Lemma 3.4.** *Any mechanism $\mathcal{M}$ satisfying weak budget-balance cannot have $\mathsf{Reciprocity}(\mathcal{M}) > 1$.*

We note that $r = 1$ implies that every NE of the mechanism reciprocates an agent exactly as much as their contribution. We refer to such 'truly fair' mechanisms as *fully reciprocal* (or just reciprocal for convenience). Lemma 3.4 shows that reciprocal mechanisms are the best one can aim for in terms of fairness among weakly budget-balanced mechanisms.

We now define a second fairness notion in the FL setting inspired by the fairness principle of equal treatment of equals (Moulin, 2002).

**Definition 3.5** (Equal treatment of equals)**.** Mechanism $\mathcal{M}$ satisfies equal treatment of equal contributors if for any sample vector $s$ and two *identical* agents $i$ and $j$, $p_i(s) = p_j(s)$.

**Efficiency metrics.** We define two natural metrics to evaluate the efficiency of a mechanism for federated learning. While fairness is a local property and can be evaluated at a solution, efficiency is a global property measuring aggregate quantities (like total payoff or data shared) and must be contrasted against an appropriate baseline. Our metrics compare the NE of a mechanism $\mathcal{M}$ against the NE of the baseline mechanism $\mathcal{M}^0$ in the absence of payments. That is, $p_i(s) = 0$ for all $i \in N$, $s \in \mathcal{S}$ for the mechanism $\mathcal{M}^0$.

The first metric, called DataGain, compares the total quantity of data shared in the worst NE of $\mathcal{M}$ to the total quantity of data shared in the best NE of the baseline $\mathcal{M}^0$.

**Definition 3.6** (Data Gain of a mechanism)**.** The data gain of a mechanism $\mathcal{M}$ is defined as:

$$\mathsf{DataGain}(\mathcal{M}) = \frac{\min\limits_{s \in \mathsf{NE}(\mathcal{M})} \|s\|_1}{\max\limits_{s^0 \in \mathsf{NE}(\mathcal{M}^0)} \|s^0\|_1}. \quad (5)$$

In a similar spirit, we define a metric AccGain to compare the cumulative payoff/accuracy of agents in a mechanism as compared to the baseline.

**Definition 3.7** (Accuracy Gain of a mechanism)**.** The accuracy gain of a mechanism $\mathcal{M}$ is defined as:

$$\mathsf{AccGain}(\mathcal{M}) = \frac{\min\limits_{s \in \mathsf{NE}(\mathcal{M})} \sum_{i=1}^n a_i(s)}{\max\limits_{s^0 \in \mathsf{NE}(\mathcal{M}^0)} \sum_{i=1}^n a_i(s^0)}. \quad (6)$$

## 4. $\mathcal{M}^{\mathsf{Shap}}$: A Fair and Efficient FL Mechanism

Recall from Example 3 that NE of $\mathcal{M}^0$ in the absence of payments can force a single agent to contribute data samples while other agents are free-riding. Therefore, we seek mechanisms that admit a NE and are both fair and efficient. Towards this goal, we present a mechanism $\mathcal{M}^{\mathsf{Shap}}$ based on the Shapley value of FL.

**Definition 4.1** (Mechanism $\mathcal{M}^{\mathsf{Shap}}$)**.** $\mathcal{M}^{\mathsf{Shap}}$ is the mechanism with the payment scheme $p$ given by

$$p_i(s) = \varphi_i^A(s) - a_i(s), \quad \text{for any } i \in N, s \in \mathcal{S}. \quad (7)$$

Note that $\varphi_i^A(s)$ is the contribution of agent $i$ towards the cumulative payoff of the agents, while $a_i(s)$ is her payoff at the sample vector $s$. Therefore, the above payment scheme can be interpreted as a *fair compensation* scheme: if an agent $i$ contributes more than what she gets as a payoff, then she is compensated via a subsidy; conversely, if her contribution is less than her payoff, she is charged via a tax.

**Cost-agnostic nature of $\mathcal{M}^{\mathsf{Shap}}$.** In contrast to the mechanisms in (Karimireddy et al., 2022; Murhekar et al., 2023), $\mathcal{M}^{\mathsf{Shap}}$ is *cost-agnostic* since it does not require knowledge of the agent cost functions. This feature offers a practical advantage: the central server (or other agents) does not need knowledge of an agent's cost function, which can sometimes prove difficult for third-parties to estimate or verify

in practice. Moreover, since the payment only depends on the actual contribution of agents to the cumulative payoff, no agent can gain by misreporting their cost functions, i.e., claiming they have a higher data collection cost than actual.

Mechanisms that attempt to compensate agents with high acquisition costs and low data quality (e.g. the mechanism of (Murhekar et al., 2023)) may unfairly penalize agents with high data quality and lower acquisition costs. In contrast, our mechanism $\mathcal{M}^{\mathsf{Shap}}$ ensures that high-quality, high-acquisition-cost data is still incentivized for sharing, as the agent's payment will compensate their data sharing costs. However, $\mathcal{M}^{\mathsf{Shap}}$ will not incentive agents with low data quality and high acquisition costs to share their data.

By construction, the server operates on a no profit or loss in our mechanism $\mathcal{M}^{\mathsf{Shap}}$ (proof in Appendix D).

**Lemma 4.2.** *The mechanism $\mathcal{M}^{\mathsf{Shap}}$ is budget-balanced.*

### 4.1. Nash equilibria of $\mathcal{M}^{\mathsf{Shap}}$: Existence

We now prove that $\mathcal{M}^{\mathsf{Shap}}$ admits a Nash equilibrium under standard assumptions on the utility functions as in Sec. 2.1.

**Theorem 4.3.** *In any federated learning instance where for every agent $i \in N$ the payoff function $a_i(\boldsymbol{s})$ is concave in $\boldsymbol{s}$, and cost function $c_i$ is non-decreasing and convex in $s_i$, the mechanism $\mathcal{M}^{\mathsf{Shap}}$ admits a Nash equilibrium.*

*Proof Sketch.* The key idea is to prove that for a fixed $\boldsymbol{s}_{-i}$, $u_i(s_i, \boldsymbol{s}_{-i}) = \varphi_i^A(s_i, \boldsymbol{s}_{-i}) - c_i(s_i)$ is concave in $s_i$. This uses the concavity of payoffs $a_i$ and the convexity of costs $c_i$. Therefore, the best response set $f_i(\boldsymbol{s}_{-i})$ is a non-empty interval, and hence that $f$ is convex valued. Lastly, the continuity of $u_i$ in $\boldsymbol{s}$ implies the upper semi-continuity of $f$. Then Kakutani's fixed point theorem implies that $f$ admits a fixed point, which then corresponds to a NE of $\mathcal{M}^{\mathsf{Shap}}$. The detailed proof can be viewed in Appendix D.

**Implications.** Theorem 4.3 shows that our mechanism $\mathcal{M}^{\mathsf{Shap}}$ *always* admits a NE, i.e., stable-state of data contribution levels, as long as agent payoffs are concave and costs are convex. These mild assumptions are commonly satisfied in practice, e.g. for all our motivating examples from Section 2.1. In the absence of this assumption, a NE is not guaranteed to exist even without any payments (Murhekar et al., 2023).

To illustrate how our mechanism circumvents free-riding by reciprocally sharing profits of collaboration, we revisit the setting of Example 3 and examine the NE under $\mathcal{M}^{\mathsf{Shap}}$. As shown below, we observe that all agents contribute data samples in the NE of $\mathcal{M}^{\mathsf{Shap}}$ as opposed to the NE of $\mathcal{M}^0$.

*Example* 4. Consider two agents with identical payoff functions $a(\boldsymbol{s}) = 1 - (\|\boldsymbol{s}\|_1 + 1)^{-1}$, and linear costs given by $c_1(s_1) = 0.1s_1$ and $c_2(s_2) = 0.25s_2$. Recall that the NE of $\mathcal{M}^0$ is given by $\boldsymbol{s}^0 = (2.16, 0)$ where agent 2 free-rides.

On the other hand, the NE of $\mathcal{M}^{\mathsf{Shap}}$ is $\boldsymbol{s}^* = (3, 1.17)$. Thus both agents contribute at the NE in $\mathcal{M}^{\mathsf{Shap}}$ and in fact $\boldsymbol{s}^*$ Pareto-dominates $\boldsymbol{s}^0$. Moreover, almost twice the amount of data is shared in the NE of $\mathcal{M}^{\mathsf{Shap}}$ as compared to $\mathcal{M}^0$.

### 4.2. Nash equilibria of $\mathcal{M}^{\mathsf{Shap}}$: Best-Response Dynamics

We next turn to the question of computing the NE of $\mathcal{M}^{\mathsf{Shap}}$. For FL instances with strongly concave utility functions, it is known from prior work (Murhekar et al., 2023) that NE can be computed by following intuitive *best-response (BR) dynamics*. The dynamics begins with an initial sample vector $\boldsymbol{s}^0$. In step $t$ with sample vector $\boldsymbol{s}^t$, each agent $i$ updates her sample contribution proportional to the gradient $\nabla_i u_i(\boldsymbol{s}^t)$ in the direction of increasing utility, while ensuring that the resulting vector $\boldsymbol{s}^{t+1}$ lies in the feasible region $\mathcal{S}$. Formally, for a step size $\delta^t > 0$, the update in step $t$ is:

$$\boldsymbol{s}^{t+1} = \boldsymbol{s}^t + \delta^t \cdot g(\boldsymbol{s}^t, \boldsymbol{\mu}^t), \tag{8}$$

where $g(\boldsymbol{s}^t, \boldsymbol{\mu}^t) := \nabla u(\boldsymbol{s}^t) + \boldsymbol{\mu}^t$ for a specific choice of vector $\boldsymbol{\mu}^t$ which ensures that $\boldsymbol{s}^{t+1} \in \mathcal{S}$, given by:

$$\mu_i^t = \begin{cases} -\nabla_i u_i(\boldsymbol{s}^t) - \frac{s_i^t}{\delta^t}, & \text{if } s_i^t + \delta^t \cdot \nabla_i u_i(\boldsymbol{s}^t) < 0 \\ -\nabla_i u_i(\boldsymbol{s}^t) + \frac{\tau_i - s_i^t}{\delta^t}, & \text{if } s_i^t + \delta^t \cdot \nabla_i u_i(\boldsymbol{s}^t) > \tau_i, \\ 0, & \text{otherwise.} \end{cases} \tag{9}$$

(Murhekar et al., 2023) show that for strongly concave utilities, the above BR dynamics converges to an $\varepsilon$-approximate NE in $O(\log(\varepsilon^{-1}))$ iterations.

In this section, we prove a stronger result: the convergence of *stochastic best-response dynamics* when utility functions are strongly concave. In stochastic BR, in each step $t$, we randomly sample a set $R^t$ of size $k$ and perform BR dynamics only for agents in $R^t$, for some fixed $k \in [n]$. With $g(\boldsymbol{s}^t, \boldsymbol{\mu}^t)$ defined as above, we define $f(\boldsymbol{s}^t, \boldsymbol{\mu}^t, R^t) \in \mathbb{R}^n$ as the random vector given by:

$$f(\boldsymbol{s}^t, \boldsymbol{\mu}^t, R^t)_i = \begin{cases} g(\boldsymbol{s}^t, \boldsymbol{\mu}^t)_i & \text{if } i \in R^t, \\ 0, & \text{otherwise.} \end{cases} \tag{10}$$

Then, for a step size $\delta^t$, the stochastic BR update is:

$$\boldsymbol{s}^{t+1} = \boldsymbol{s}^t + \delta^t \cdot f(\boldsymbol{s}^t, \boldsymbol{\mu}^t, R^t). \tag{11}$$

**Theorem 4.4.** *For a concave game where agent utility functions are (i) $\lambda$-strongly concave: $(G + \lambda \cdot I_{n \times n})$ is negative semi-definite, and (ii) $L$-bounded derivatives: $|G_{ij}| \leq L$, for constants $\lambda, L > 0$, stochastic best response dynamics (11) with step size $\delta^t = \frac{n-1}{k-1} \cdot \frac{\lambda}{n^2 L^2}$ converges to an approximate Nash equilibrium $\boldsymbol{s}^T$ where $\|g(\boldsymbol{s}^T, \boldsymbol{\mu}^T)\|_2 < \varepsilon$ in $T$ iterations, where:*

$$T = \frac{2n^2 L^2}{\lambda^2} \log\left(\frac{\|g(\boldsymbol{s}^0, \boldsymbol{\mu}^0)\|_2}{\varepsilon}\right).$$

The proof is deferred to App. D. Finally, we show that $\mathcal{M}^{\mathsf{Shap}}$ does satisfy conditions of Theorem 4.4, proving that its NE can be computed via stochastic BR dynamics.

**Lemma 4.5.** *For a federated learning instance where agent (i) payoff functions are $\lambda_1$-strongly concave and cost functions are $\lambda_2$-strongly concave, and (ii) second derivatives of payoffs and costs are bounded: $|\frac{\partial^2 a_i}{\partial s_j \partial s_k}| \leq L_1$ and $|\frac{\partial^2 c_i}{\partial^2 s_i}| \leq L_2$, for constants $\lambda_1, \lambda_2, L_1, L_2 > 0$, stochastic best response dynamics (11) with step size $\delta^t = \frac{n-1}{k-1} \cdot \frac{n\lambda_1 + \lambda_2}{n^2(2nL_1 + L_2)^2}$ converges to an approximate Nash equilibrium $\boldsymbol{s}^T$ where $\|g(\boldsymbol{s}^T, \boldsymbol{\mu}^T)\|_2 < \varepsilon$ in $T$ iterations, where:*

$$T = \frac{2n^2 \cdot (2nL_1 + L_2)^2}{(n\lambda_1 + \lambda_2)^2} \log\left(\frac{\|g(\boldsymbol{s}^0, \boldsymbol{\mu}^0)\|_2}{\varepsilon}\right)$$

**Implications of Theorem 4.4 and Lemma 4.5.** The above results show that stochastic BR dynamics convergences for strongly concave games in $O(\log(\varepsilon^{-1}))$ iterations with an $\varepsilon$-approximate NE. Indeed, we employ stochastic BR dynamics to compute NE of $\mathcal{M}^{\mathsf{Shap}}$ in our experiments (see Section 5.2). Note that the payoff and cost functions defined by generalization bounds (Equation (2)) and those observed in practice (Equation (12)) satisfy the assumptions of results.

### 4.3. Fairness and Efficiency Properties of $\mathcal{M}^{\mathsf{Shap}}$

We now establish desirable properties of our mechanism $\mathcal{M}^{\mathsf{Shap}}$, focusing on individual rationality, fairness, and efficiency. We defer proofs to Appendix D due to space constraints and highlight the key implications of our results.

**Lemma 4.6.** $\mathcal{M}^{\mathsf{Shap}}$ *is individually rational.*

This shows that no agent will receive a negative utility by participating in our mechanism. Next, we show that $\mathcal{M}^{\mathsf{Shap}}$ satisfies the fairness principles outlined in Def. 3.3 and 3.5.

**Theorem 4.7.** $\mathcal{M}^{\mathsf{Shap}}$ *satisfies* $\mathsf{Reciprocity}(\mathcal{M}^{\mathsf{Shap}}) = 1$, *i.e., is fully reciprocal. Moreover, $\mathcal{M}^{\mathsf{Shap}}$ satisfies equal treatment of equals.*

Therefore, $\mathcal{M}^{\mathsf{Shap}}$ ensures that every agent receives as much from the mechanism as their contribution to other agents.

Having shown that $\mathcal{M}^{\mathsf{Shap}}$ is reciprocally fair, we next investigate its efficiency properties in terms of data contributions, welfare, and accuracy. We first prove a general tradeoff between the data contribution and overall welfare (i.e., total accuracy minus costs) at the NE of our mechanism $\mathcal{M}^{\mathsf{Shap}}$ as compared to any other sample vector.

**Theorem 4.8.** *Let $W(\boldsymbol{s}) = A(\boldsymbol{s}) - \sum_{i \in [n]} c_i(s_i)$ denote the total welfare of the agents, and let $\boldsymbol{s}^* \in \mathsf{NE}(\mathcal{M}^{\mathsf{Shap}})$. Consider any data contribution vector $\boldsymbol{s}$ that weakly Pareto-dominates $\boldsymbol{s}^*$, i.e., $s_i \geq s_i^*$ for all $i$. Then $W(\boldsymbol{s}) < W(\boldsymbol{s}^*)$.*

**Implications.** Theorem 4.8 allows us to compare *any* sample vector $\boldsymbol{s}$ (e.g. the NE of some mechanism $\mathcal{M}$) with the NE $\boldsymbol{s}^*$ of our mechanism $\mathcal{M}^{\mathsf{Shap}}$ in terms of data contributions and utilitarian welfare. In particular:

1. If all agents share equal or strictly higher data in $\boldsymbol{s}$ than $\boldsymbol{s}^*$, then the welfare satisfies $W(\boldsymbol{s}) < W(\boldsymbol{s}^*)$. Thus, there is no mechanism $\mathcal{M}$ in whose NE agents contribute at least as much data as they would in the NE of $\mathcal{M}^{\mathsf{Shap}}$, while also achieving higher welfare.

2. Conversely, if $W(\boldsymbol{s}) \geq W(\boldsymbol{s}^*)$, then $\boldsymbol{s}$ does not weakly Pareto-dominate $\boldsymbol{s}^*$. Thus, any mechanism $\mathcal{M}$ whose NE achieves higher welfare than the NE of $\mathcal{M}^{Shap}$, will necessarily have at least one agent who contributes strictly less data at the NE of $\mathcal{M}$ than at the NE of $\mathcal{M}^{Shap}$.

Theorem 4.8 thus underscores that our reciprocal mechanism $\mathcal{M}^{\mathsf{Shap}}$ exhibits favorable properties in terms of *both* data contribution at equilibrium (in the spirit of the mechanism of (Karimireddy et al., 2022)), and welfare (in the spirit of the mechanism of (Murhekar et al., 2023)).

We further formalize this by explicitly quantifying the data gain and accuracy gain of $\mathcal{M}^{\mathsf{Shap}}$ when the accuracy functions are identical and take a tractable form, and costs are (possibly different, but) linear. Examples 1 and 2 indicate that the accuracy $a(m)$ on a learning task with $m$ data points varies as $a(m) = 1 - \alpha \cdot m^{-\beta}$ for $\alpha, \beta \in (0, 1]$. This motivates us to assume the following canonical form for the accuracy $a(\boldsymbol{s})$ of an agent $i$ as a function of the contribution vector $\boldsymbol{s}$.

$$a(\boldsymbol{s}) = 1 - \frac{\alpha}{(\|\boldsymbol{s}\|_1 + 1)^\beta}. \tag{12}$$

Note that the normalization ensures that $a(\cdot)$ is concave, and $a(\boldsymbol{s}) \in [0, 1]$ for all $\boldsymbol{s} \in \mathcal{S}$.

**Theorem 4.9.** *Consider any FL instance with $n$ agents where agents have (i) identical payoff function $a(\boldsymbol{s}) = 1 - \alpha \cdot (\|\boldsymbol{s}\|_1 + 1)^{-\beta}$ for $\alpha > 0$ and $\beta \in (0, 1]$, and (ii) linear cost functions $c_i(s_i) = \gamma_i \cdot s_i + d_i$ for $\gamma_i, d_i \geq 0$. Then $\mathcal{M}^{\mathsf{Shap}}$ satisfies:*

*(i)* $\mathsf{DataGain}(\mathcal{M}^{\mathsf{Shap}}) \geq n^{\frac{1}{\beta+1}}$, *and*

*(ii)* $\mathsf{AccGain}(\mathcal{M}^{\mathsf{Shap}}) \geq 1 + \alpha^{\frac{1}{\beta+1}} \cdot \beta^{\frac{-\beta}{\beta+1}} \cdot (\min_i \gamma_i)^{\frac{\beta}{\beta+1}} \cdot \left(1 - n^{-\frac{\beta}{1+\beta}}\right)$.

**Implications.** The above theorem establishes lower bounds on aggregate quantities (total amount of data shared and cumulative payoff) in any NE of our mechanism $\mathcal{M}^{\mathsf{Shap}}$ compared to any NE in the absence of payments. We note that the gain in total data shared is at least $n^{\frac{1}{\beta+1}}$, which is at least $\sqrt{n}$ since $\beta \in (0, 1]$. This establishes that the gain in data shared increases with the number of agents $n$

joining the federation, which is the natural expectation from mechanisms for FL. Likewise, we observe that the gain in cumulative payoff is strictly more than 1, and increases with an increasing number of agents.

The assumptions of linear costs and identical payoffs with forms similar to Eq. (12) are standard in prior works (Blum et al., 2021; Karimireddy et al., 2022; Murhekar et al., 2023). In our experiments (Sec. 5), we observe that $\mathcal{M}^{\mathsf{Shap}}$ outperforms other mechanisms in terms of data gain, accuracy gain, and reciprocity, even *without* making any assumption on payoff functions taking a particular form, e.g., (12).

## 5. Empirical Evaluation

In this section, we compare the performance of our mechanism $\mathcal{M}^{\mathsf{Shap}}$ with two baselines $M^0$ and $\mathcal{M}^{\mathsf{BG}}$ (Murhekar et al., 2023) tailored to maximize welfare. In Sec 5.1, we compare the performance of the NEs found by the best response dynamics of the three mechanisms when the accuracy and cost functions are given as closed-form functions. In Sec 5.2, we provide a practical distributed FL protocol `FedBR-Shap` to approximately find an NE.

**Datasets and local training.** We evaluate the mechanisms on five datasets, including three image-based datasets: MNIST, FashionMNIST, and CIFAR-10, a healthcare dataset (Afshari Safavi, 2021), and a synthetic dataset. A simple CNN network is used for the local training for MNIST and FashionMNIST, and ResNet-10 (He et al., 2016) is used for CIFAR-10. We use ResMLP (Touvron et al., 2022) for local training of the healthcare dataset, which contains meteorological and geospatial features related to Lumpy Skin Disease. In addition, we create a synthetic binary classification dataset with 10 input variables and perform a quadratic regression.

### 5.1. Performance of Nash Equilibria

In this part, we implement the best response dynamics (as described in Sec 4.2) for the three mechanisms and compare the performance of the NE found by best response dynamics.

**Setup.** We set the number of agents as 30 for the three image-based datasets (MNIST, FashionMNIST, and CIFAR-10) and randomly sample 10 agents to update their shares in each iteration. For the remaining two datasets, we set the number of agents to two and perform no sampling. We adopt the statistical heterogeneous setting: the datasets of the agents are Non-Independent Identically Distributed (Non-IID) (Ye et al., 2023). We partition all the agents into $T$ groups. Denote the agents in the $t$-th group by $A_t$. In the three image-based datasets, the agents are equally partitioned into three groups (i.e., $T = 3$), each of which contains 10 agents. The images of the three groups are rotated

by particular angles $10°, 90°$, and $180°$, respectively. For the other two datasets, we set the dataset of one agent to consist of $70\%$ positive data points and $30\%$ negative data points, and the other agent vice versa.

**Interpretation of sample vector $s$.** We note that, in our implementation, unlike the model described in Sec. 2, we do not ask agents to transmit data to the central server. Instead, agents perform training locally and only transmit the gradient information to the central server. Hence, the number of data samples $s_i$ is interpreted as the number of batches trained by agent $i$ during each local training.

**Closed-form accuracy and cost.** We assume the accuracy function of an agent in the $i$-th group is in the form of

$$a_i(\boldsymbol{s}) = 1 - \frac{1}{1 + \sum_{t=1}^{T} w_{i,t} \cdot \sum_{j \in A_t} s_j}, \quad i \in [T].$$

Before running the best-response dynamics, we randomly sample a number of data points from each group and perform a normal federated training without strategic behaviors. After collecting a set of accuracy results, we fit the accuracy functions using the non-linear least squares method. After that, we assume the central server broadcasts all the parameters $\{w_{i,t}\}_{i,t \in [T]}$ to all agents. Meanwhile, each agent is assumed to incur a linear cost: $c_i(\boldsymbol{s}) = \gamma_i \cdot s_i$, where $\gamma_i$ is set randomly in advance.

Based on the closed-form utility function, each agent can (approximately) compute the derivatives of the Shapley share at a specific sample vector $\boldsymbol{s}$. Note that our mechanism requires that all agents operate with a synchronized sample vector. In a cross-silo FL setting, where the number of participating clients is relatively small, this synchronization overhead is practical. For the case of $n = 30$ agents, the agents perform the Monte-Carlo approximation by randomly sampling $\lfloor n \log n \rfloor = 102$ permutations and computing the average. In contrast, for two agents, we directly compute the exact derivative of the Shapley value. We duplicate each experiment 3 times, and the appropriateness of the number of sampled permutations is supported by the low standard deviation of the reported results.

**Experimental results.** Table 1 reports the performance of the NEs of $\mathcal{M}^{\mathsf{Shap}}$ and the two baseline mechanisms. The tables report the model accuracy, welfare, data gain, and the reciprocity of the mechanisms. The "Avg Accuracy" column reports the average accuracy of the final model evaluated on all agents' test datasets. It is worth noting that Table 1 reports the *actual accuracies* of the models trained using the data contributions at NE. The "Data Share" column shows the ratio of shared data to total data. For the two columns "Data Gain" and "Accuracy Gain", we report the relative

*Table 1.* Performance of the NEs of $\mathcal{M}^{\mathsf{Shap}}$ and the two baselines $\mathcal{M}^0$ and $\mathcal{M}^{\mathsf{BG}}$.

| Dataset | Avg. Accuracy (%) | | | Data Share (%) | | | Data Gain | | | Accuracy Gain | | | Reciprocity | | |
|---|---|---|---|---|---|---|---|---|---|---|---|---|---|---|---|
| | $\mathcal{M}^0$ | $\mathcal{M}^{\mathsf{BG}}$ | $\mathcal{M}^{\mathsf{Shap}}$ | $\mathcal{M}^0$ | $\mathcal{M}^{\mathsf{BG}}$ | $\mathcal{M}^{\mathsf{Shap}}$ | $\mathcal{M}^0$ | $\mathcal{M}^{\mathsf{BG}}$ | $\mathcal{M}^{\mathsf{Shap}}$ | $\mathcal{M}^0$ | $\mathcal{M}^{\mathsf{BG}}$ | $\mathcal{M}^{\mathsf{Shap}}$ | $\mathcal{M}^0$ | $\mathcal{M}^{\mathsf{BG}}$ | $\mathcal{M}^{\mathsf{Shap}}$ |
| MNIST | 88.3 | 87.6 | **90.4** | 5.8 | 7.6 | **54.9** | 1.00 | 1.32 | **9.51** | 1.00 | 0.99 | **1.02** | 0.61 | 0.70 | **1.00** |
| FashionMNIST | 60.9 | 61.5 | **63.3** | 4.1 | 6.2 | **54.8** | 1.00 | 1.51 | **13.43** | 1.00 | 1.02 | **1.05** | 0.47 | 0.75 | **1.00** |
| CIFAR-10 | 43.6 | 44.7 | **48.5** | 25.6 | 28.9 | **99.6** | 1.00 | 1.13 | **3.89** | 1.00 | 1.02 | **1.11** | 0.50 | 0.57 | **1.00** |
| Lumpy-Skin-Disease | 94.2 | 94.0 | **95.2** | 46.7 | 46.7 | **81.3** | 1.00 | 1.00 | **1.73** | 1.00 | 0.98 | **1.01** | 0.92 | 0.02 | **1.00** |
| Quadratic | 67.2 | 65.8 | **90.8** | 3.3 | 4.0 | **99.6** | 1.00 | 1.25 | **31.18** | 1.00 | 0.98 | **1.36** | 0.93 | 0.95 | **1.00** |

values of data share and accuracy compared to the baseline mechanism $\mathcal{M}^0$.

Our results demonstrate that our mechanism $\mathcal{M}^{\mathsf{Shap}}$ outperforms baselines in terms of both fairness and efficiency.

- $\mathcal{M}^{\mathsf{Shap}}$ significantly incentivizes agents to contribute more data compared to both baselines, as evidenced by a $9.51\times$ data gain for MNIST, a $13.43\times$ data gain for FashionMNIST, and a $3.89\times$ data gain for the CIFAR-10. FedBR-Shap also outperforms baselines in terms of accuracy, reporting an accuracy gain of $1.02\times$ to $1.11\times$ for the three image-based datasets. The accuracy gain is extremely high ($1.36\times$) in the synthetic dataset.

- We observe that the reciprocity of the baselines is strictly lower than 1, indicating that they force some agents to contribute more to the federation than what they receive. $\mathcal{M}^{\mathsf{Shap}}$ is fairer as it is always fully reciprocal.

- Our mechanism $\mathcal{M}^{\mathsf{Shap}}$ guarantees reciprocal fairness in more practically meaningful scenarios. In the skin disease prediction task, the first agent possesses a large number of positive (i.e., disease-present) patient samples, which are expected to be more informative for the prediction task. The fitted accuracy functions in Sec E also reflect this.

  Observe that, in the NEs of the two baselines, agent 1 even needs to share more data than the second agent (who possesses lower-quality data): the NEs in $\mathcal{M}^0$ and $\mathcal{M}^{\mathsf{BG}}$ are respectively $(30.5\%, 18.1\%)$ and $(30.6\%, 18.1\%)$. In contrast, in $\mathcal{M}^{\mathsf{Shap}}$, the NE is $(38.1\%, 55.4\%)$, indicating that agent 2 should contribute more than agent 1 in the NE. The finding also highlights the reciprocal fairness ensured by $\mathcal{M}^{\mathsf{Shap}}$.

### 5.2. FedBR-Shap: FL Protocol for $\mathcal{M}^{\mathsf{Shap}}$

Note that the above computation of NE relies on known closed-form accuracy functions. Unfortunately, a closed-form accuracy function may not always be possible, especially when the information about the agents' dataset is inaccessible. Moreover, another restriction of the above implementation is assuming agents are always training on fixed number of batches throughout the learning process. However, an agent may not always follow a static strategy in real training and can adjust it strategically based on the current

model. For example, an agent may skip local training if the current model is already good enough for her.

To address the limitations mentioned above, we design a distributed FL protocol, FedBR-Shap, which implements $\mathcal{M}^{\mathsf{Shap}}$ in practice and approximately computes NE in more realistic scenarios. FedBR-Shap computes NE while simultaneously training a global model for the underlying FL task. FedBR-Shap runs for $T$ iterations. In each iteration, the central server updates the global model, and a set of agents updates their sample vectors according to the transition relation (Equation (8)) of best response dynamics. The update differs from the previous approaches as follows:

- The central server gives payments to the agents at the end of every iteration. Each agent $i$ is allowed to change the number of samples $s_i$ during the training. The payment follows $\mathcal{M}^{\mathsf{Shap}}$ according to their contribution to the accuracy improvement for the current iteration.

- There are no publicly known closed-form accuracy functions. Instead, at iteration $t$, each agent $i \in [n]$ locally trains two models $T_i$ and $T_i^\epsilon$ using $s_i^t$ and $s_i^t + \epsilon$ batches of its local dataset. The central server updates its global model $\theta^{t-1}$ to $\theta^t$ using all the $T_i$. In addition, it aggregates a set of intermediate models using subsets of $T_i$ and $T_i^\epsilon$. The central server then distributes the intermediate models to the agents, which enables them to approximately estimate the derivative of the utility function and update their samples correspondingly.

We emphasize that FedBR-Shap is a truly distributed protocol for FL, as it does not involve actual data sharing, only relies on the gradients of each agent's local datasets, and does not make any assumptions about accuracy functions obeying a closed-form expression. This is unlike prior implementations of mechanisms for FL (e.g., Sec. 5.1, (Karimireddy et al., 2022; Murhekar et al., 2023)). Moreover, FedBR-Shap does not merely calculate a single NE. We take an observation window size $W$ and treat the sample vector collected over every $W$ iterations as the NE for that stage. We defer the complete description and pseudo-codes of FedBR-Shap to Appendix F.

## Acknowledgements

The research of Bhaskar Ray Chaudhury is supported by the NSF CAREER grant CCF No. 2441580. The research of Aniket Murhekar and Ruta Mehta is supported by NSF grant CCF 2334461.

## Impact Statement

The goal of this paper is to advance *fairness* in machine learning, specifically in the context of federated learning. Potential societal consequences of our work include the design of fairer federated learning protocols. Since fairness is often a subjective notion, it is conceivable for our fairness notion of reciprocity to conflict with other desiderata.

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

## A. Discussion and Conclusion

Our goal in this work is to address issues of fairness, efficiency, and data-sharing incentives faced by agents participating in federated learning who incur data-sharing costs. We propose a budget-balanced, reciprocally fair mechanism for federated learning, in which agents are incentivized via payments that reflect their 'contribution' to the federation. We defined natural metrics called data gain and accuracy gain to measure the efficiency of mechanisms for FL. We proved that our mechanism achieves significant gains for specialized forms of utility functions and substantiated this empirically.

Our work leaves several directions for future work. The first direction is establishing theoretical lower bounds on the data gain for a wider class of utility functions. Another direction is to further explore the trade-off between fairness and efficiency, by designing $r$-reciprocal mechanisms for $r < 1$ that achieve provably higher data gain. Lastly, investigating other profit-sharing methods instead of the Shapley share is another direction towards fair mechanism design for FL.

## B. Examples

*Example* 5 (Linear or Random discovery (Blum et al., 2021)). Consider a setting where each agent has a sampling probability distribution $\mathbf{q}_i$ over a given instance space $X$ and gets a reward equalling $q_{ix}$ whenever the instance $x$ is sampled by any agent. Then the expected payoff to agent $i$ is $a_i(\boldsymbol{s}) = (QQ^T\boldsymbol{s})_i$, where $Q$ is a matrix given by $Q[i, x] = q_{ix}$ for $i \in N$ and $x \in X$. Here $W = QQ^T$ is a symmetric PSD matrix with an all-one diagonal. Thus in this model, the payoff is linear in the sample vector and is given by $a_i(\boldsymbol{s}) = (W\boldsymbol{s})_i$ for a matrix $W$.

*Example* 6 (Random coverage (Blum et al., 2021)). Consider a modification of the setting in Example 5 where agent $i$ obtains the reward $q_{ix}$ only once if $x$ is sampled. Thus in this model, the payoff given by expected accuracy takes the form:

$$a_i(\boldsymbol{s}) = 1 - \frac{1}{2} \sum_{x \in X} q_{ix} \prod_{j=1}^{n} (1 - q_{jx})^{s_j} \in [0, 1]. \tag{13}$$

*Example* 7. Consider two agents with identical payoff functions $a(\boldsymbol{s}) = 1 - (\|\boldsymbol{s}\|_1 + 1)^{-1}$, and linear cost functions given by $c_1(s_1) = 0.1s_1$ and $c_2(s_2) = 0.25s_2$. Independently, the optimal contributions of the agents are $s_1^0 = 2.16$ and $s_2^0 = 1$ respectively.

According to the mechanism of (Karimireddy et al., 2022), the NE contribution $(s_1, s_2)$ satisfies:

$$1 - \frac{1}{s_1 + s_2 + 1} = 1 - \frac{1}{s_1^0 + 1} + (0.1 + \varepsilon) \cdot (s_1 - s_1^0),$$

$$1 - \frac{1}{s_1 + s_2 + 1} = 1 - \frac{1}{s_2^0 + 1} + (0.25 + \varepsilon) \cdot (s_2 - s_2^0),$$

where $\varepsilon \to 0$. This leads to an NE given by $\boldsymbol{s} = (3.98, 2.46)$. At $\boldsymbol{s}$, the Shapley shares are $\varphi_1^A(\boldsymbol{s}) = 0.9538$ and $\varphi_2^A(\boldsymbol{s}) = 0.7772$, while the payoff is $a(\boldsymbol{s}) = 0.8656$. This the reciprocity of the mechanism is $\frac{0.8656}{0.9538} = 0.9 < 1$, i.e., the mechanism forces agent 1 to contribute more than she gets from the mechanism.

## C. Proofs from Section 2

**Lemma 2.3.** *For any $\boldsymbol{s} \in \mathcal{S}$, $A(\boldsymbol{s}) = \sum_{i \in N} \varphi_i^A(\boldsymbol{s})$.*

*Proof.* Consider the summation of $\varphi_i^A(\boldsymbol{s})$. For each subset $X \subseteq [n]$, if $X$ is non-empty and $X \neq [n]$, $A(\boldsymbol{s}[X])$ occurs in the first terms of all the $\phi_i^A(\boldsymbol{s})$ for $|X|$ times. Besides, it occurs in the second terms of all the $\phi_i^A(\boldsymbol{s})$ for $n - |X|$ times. Hence, the coefficient of $A(\boldsymbol{s}[X])$ is given by

$$\frac{1}{n} \cdot \left( |X| \cdot \binom{n-1}{|X|-1}^{-1} - (n - |X|) \cdot \binom{n-1}{|X|}^{-1} \right),$$

which is equal to zero. Besides, when $X = [n]$, it only occurs at the second terms of $\varphi_i^A(\boldsymbol{s})$. The coefficient of $A(\boldsymbol{s}[[n]]) = A(\boldsymbol{s})$ is given by $\frac{1}{n} \cdot n \cdot \binom{n-1}{n-1}^{-1} = 1$. Additionally, since $A(\boldsymbol{s}[\emptyset]) = 0$, the summation of $\varphi_i^A(\boldsymbol{s})$ is equal to $A(\boldsymbol{s})$. $\square$

# D. Proofs from Section 4

**Theorem 4.3.** *In any federated learning instance where for every agent $i \in N$ the payoff function $a_i(\boldsymbol{s})$ is concave in $\boldsymbol{s}$, and cost function $c_i$ is non-decreasing and convex in $s_i$, the mechanism $\mathcal{M}^{\mathsf{Shap}}$ admits a Nash equilibrium.*

*Proof.* We consider the best response correspondence $f$ of agents under the mechanism $\mathcal{M}^{\mathsf{Shap}}$. Thus, $f$ is the correspondence given by:

$$f_i(\boldsymbol{s}_{-i}) = \arg\max_{x \in S_i}\{a_i(x, \boldsymbol{s}_{-i}) - c_i(x) + p_i(\boldsymbol{s})\} = \arg\max_{x \in S_i}\{\varphi^A(x, \boldsymbol{s}_{-i}) - c_i(x)\},$$

for all $i \in N$, where the last equality used the payment rule of $\mathcal{M}^{\mathsf{Shap}}$ given by (7). Using Proposition 2.2, we prove the existence of a Nash equilibrium by showing that $f$ has a fixed point.

To this end, we first note that $f$ is defined over a compact, convex domain $\mathcal{S}$. Further, the continuity of $a_i$ and $c_i$ in $\boldsymbol{s}$ implies that $u_i(\boldsymbol{s}) := \varphi_i^A(\boldsymbol{s}) - c_i(s_i)$ is continuous in $\boldsymbol{s}$ for each agent $i$. We now show that for fixed $\boldsymbol{s}_{-i}$, $u_i(s_i, \boldsymbol{s}_{-i})$ is concave in $s_i$. Observe that:

$$\frac{\partial^2 u_i}{\partial s_i^2} = \frac{\partial^2 \varphi_i^A(\boldsymbol{s})}{\partial s_i^2} - \frac{\partial^2 c_i}{\partial s_i^2}$$

$$= \frac{1}{n} \cdot \sum_{X \subseteq N \setminus \{i\}} \binom{n-1}{|X|}^{-1} \cdot \left(\frac{\partial^2 A(\boldsymbol{s}[X \cup \{i\}])}{\partial s_i^2} - \frac{\partial^2 A(\boldsymbol{s}[X])}{\partial s_i^2}\right) - \frac{\partial^2 c_i}{\partial s_i^2}$$

$$= \frac{1}{n} \cdot \sum_{X \subseteq N \setminus \{i\}} \binom{n-1}{|X|}^{-1} \cdot \left(\frac{\partial^2 A(\boldsymbol{s}[X \cup \{i\}])}{\partial s_i^2}\right) - \frac{\partial^2 c_i}{\partial s_i^2} \qquad \text{(since } i \notin X\text{)}$$

$$< 0,$$

where the last step used the fact that $A = \sum_i a_i$ is concave in $s_i$ and $c_i$ is convex in $s_i$. Since $\frac{\partial^2 u_i}{\partial s_i^2} < 0$, we conclude that $u_i$ is concave in $s_i$ for any fixed $\boldsymbol{s}_{-i}$. This implies that the best response set $f_i(\boldsymbol{s}_{-i})$ is a non-empty interval, and hence that $f$ is convex valued. Lastly, the continuity of $u_i$ in $\boldsymbol{s}$ implies that $f$ is upper semi-continuous.

Kakutani's fixed-point theorem states (Kakutani, 1941) that every upper semi-continuous non-empty and convex valued correspondence defined over a compact, convex domain admits a fixed point. Since we argued above that $f$ satisfies the conditions of Kakutani's fixed point theorem, we conclude that $f$ admits a fixed point, and hence that $\mathcal{M}^{\mathsf{Shap}}$ admits a Nash equilibrium. $\square$

**Lemma 4.2.** *The mechanism $\mathcal{M}^{\mathsf{Shap}}$ is budget-balanced.*

*Proof.* At any sample vector $\boldsymbol{s}$, $\sum_i p_i(\boldsymbol{s}) = \sum_i \varphi_i^A(\boldsymbol{s}) - \sum_i a_i(\boldsymbol{s}) = 0$, using Lemma 2.3. $\square$

**Theorem 4.4.** *For a concave game where agent utility functions are (i) $\lambda$-strongly concave: $(G + \lambda \cdot I_{n \times n})$ is negative semi-definite, and (ii) $L$-bounded derivatives: $|G_{ij}| \leq L$, for constants $\lambda, L > 0$, stochastic best response dynamics (11) with step size $\delta^t = \frac{n-1}{k-1} \cdot \frac{\lambda}{n^2 L^2}$ converges to an approximate Nash equilibrium $\boldsymbol{s}^T$ where $\|g(\boldsymbol{s}^T, \boldsymbol{\mu}^T)\|_2 < \varepsilon$ in $T$ iterations, where:*

$$T = \frac{2n^2 L^2}{\lambda^2} \log\left(\frac{\|g(\boldsymbol{s}^0, \boldsymbol{\mu}^0)\|_2}{\varepsilon}\right).$$

*Proof.* Our goal is to compare $\|g(\boldsymbol{s}^{t+1}, \boldsymbol{\mu}^{t+1})\|_2$ and $\|g(\boldsymbol{s}^t, \boldsymbol{\mu}^t)\|_2$. To do this, we first observe that by the definition of $\boldsymbol{\mu}^{t+1}$ from (9), we have for every $i \in [n]$, $|g(\boldsymbol{s}^{t+1}, \boldsymbol{\mu}^{t+1})_i| \leq |g(\boldsymbol{s}^{t+1}, \boldsymbol{\mu}^t)_i|$. This implies:

$$\|g(\boldsymbol{s}^{t+1}, \boldsymbol{\mu}^{t+1})\|_2 \leq \|g(\boldsymbol{s}^{t+1}, \boldsymbol{\mu}^t)\|_2 \tag{14}$$

Next, we compare $\|g(\boldsymbol{s}^{t+1}, \boldsymbol{\mu}^t)\|_2$ and $\|g(\boldsymbol{s}^t, \boldsymbol{\mu}^t)\|_2$. Observe that:

$$g(\boldsymbol{s}^{t+1}, \boldsymbol{\mu}^t) - g(\boldsymbol{s}^t, \boldsymbol{\mu}^t) = (\nabla u(\boldsymbol{s}^{t+1}) + \boldsymbol{\mu}^t) - (\nabla u(\boldsymbol{s}^t) + \boldsymbol{\mu}^t)$$
$$= (\nabla u(\boldsymbol{s}^{t+1}) - \nabla u(\boldsymbol{s}^t)). \tag{15}$$

Using Taylor's expansion, we have:

$$\nabla u(\boldsymbol{s}^{t+1}) - \nabla u(\boldsymbol{s}^t) = G(\boldsymbol{s}') \cdot (\boldsymbol{s}^{t+1} - \boldsymbol{s}^t),$$

where $\boldsymbol{s}' = \boldsymbol{s}^t + \alpha(\boldsymbol{s}^{t+1} - \boldsymbol{s}^t)$ for some $\alpha \in [0, 1]$ and $G$ is the Jacobian of $\nabla u$. Using Equation (15), we get:

$$g(\boldsymbol{s}^{t+1}, \boldsymbol{\mu}^t) = g(\boldsymbol{s}^t, \boldsymbol{\mu}^t) + G(\boldsymbol{s}') \cdot (\boldsymbol{s}^{t+1} - \boldsymbol{s}^t). \tag{16}$$

To bound the $\|g(\boldsymbol{s}^{t+1}, \boldsymbol{\mu}^t)\|_2$ in terms of $\|g(\boldsymbol{s}^t, \boldsymbol{\mu}^t)\|_2$ using the above equation, we will use the stochastic BR dynamics update rule Equation (11) and relate $g(\boldsymbol{s}^t, \boldsymbol{\mu}^t)$ with $f(\boldsymbol{s}^t, \boldsymbol{\mu}^t, R^t)$.

To this end, let $D$ denote the uniform distribution over all subsets of size $k$ drawn from the set of agents. By the definition of Equation (10), we have:

$$\mathbb{E}_{R^t \sim D}[f(\boldsymbol{s}^t, \boldsymbol{\mu}^t, R^t)] = \frac{k}{n} \cdot g(\boldsymbol{s}^t, \boldsymbol{\mu}^t). \tag{17}$$

Using Equation (14), we have $\|f(\boldsymbol{s}^{t+1}, \boldsymbol{\mu}^{t+1}, R^{t+1})\|_2 \leq \|f(\boldsymbol{s}^{t+1}, \boldsymbol{\mu}^t, R^{t+1})\|_2$ for all $R^{t+1}$. This implies:

$$
\begin{aligned}
\mathbb{E}_{R^{t+1} \sim D}[\|f(\boldsymbol{s}^{t+1}, \boldsymbol{\mu}^{t+1}, R^{t+1})\|_2] &\leq \mathbb{E}_{R^{t+1} \sim D}[\|f(\boldsymbol{s}^{t+1}, \boldsymbol{\mu}^t, R^{t+1})\|_2] \\
&= \mathbb{E}_{R^t \sim D}[\|f(\boldsymbol{s}^{t+1}, \boldsymbol{\mu}^t, R^t)\|_2],
\end{aligned}
\tag{18}
$$

where the equality used the fact that the sets $R^t$ and $R^{t+1}$ are sampled from the same distribution $D$.

We now relate $f(\boldsymbol{s}^{t+1}, \boldsymbol{\mu}^t, R^t)$ and $f(\boldsymbol{s}^t, \boldsymbol{\mu}^t, R^t)$ using (16):

$$f(\boldsymbol{s}^{t+1}, \boldsymbol{\mu}^t, R^t) = f(\boldsymbol{s}^t, \boldsymbol{\mu}^t, R^t) + I_{R^t} \cdot G(\boldsymbol{s}') \cdot (\boldsymbol{s}^{t+1} - \boldsymbol{s}^t),$$

where $I_{R^t} \in \{0, 1\}^{n \times n}$ is the diagonal matrix with $I_{R^t}[i, i] = 1$ iff $i \in R^t$. Using the update equation (11), we have:

$$f(\boldsymbol{s}^{t+1}, \boldsymbol{\mu}^t, R^t) = f(\boldsymbol{s}^t, \boldsymbol{\mu}^t, R^t) + \delta^t \cdot I_{R^t} \cdot G(\boldsymbol{s}') \cdot f(\boldsymbol{s}^t, \boldsymbol{\mu}^t, R^t). \tag{19}$$

Let us next evaluate the expected value of the rightmost term. Fix $i \in [n]$.

$$
\begin{aligned}
\mathbb{E}[(I_{R^t} \cdot G(\boldsymbol{s}') \cdot f(\boldsymbol{s}^t, \boldsymbol{\mu}^t, R^t))_i] &= \sum_j (I_{R^t} \cdot G(\boldsymbol{s}'))[i, j] \cdot f(\boldsymbol{s}^t, \boldsymbol{\mu}^t, R^t)_j \\
&= \Pr[i \in R^t] \cdot \sum_j \Pr[j \in R^t \mid i \in R^t] \cdot G(\boldsymbol{s}')[i, j] \cdot g(\boldsymbol{s}^t, \boldsymbol{\mu}^t)_j \\
&= \frac{k}{n} \cdot \frac{k-1}{n-1} \sum_j G(\boldsymbol{s}')[i, j] \cdot g(\boldsymbol{s}^t, \boldsymbol{\mu}^t)_j \\
&= \frac{k}{n} \cdot \frac{k-1}{n-1} \cdot (G(\boldsymbol{s}') \cdot g(\boldsymbol{s}^t, \boldsymbol{\mu}^t))_i.
\end{aligned}
\tag{20}
$$

Taking the expectation of (19) and using the above equality, we get:

$$\mathbb{E}_{R^t \sim D}[f(\boldsymbol{s}^{t+1}, \boldsymbol{\mu}^t, R^t)] = \mathbb{E}_{R^t \sim D}[f(\boldsymbol{s}^t, \boldsymbol{\mu}^t, R^t)] + \delta^t \cdot \frac{k}{n} \cdot \frac{k-1}{n-1} \cdot (G(\boldsymbol{s}') \cdot g(\boldsymbol{s}^t, \boldsymbol{\mu}^t))$$

Using (17) in the above, we get:

$$g(\boldsymbol{s}^{t+1}, \boldsymbol{\mu}^t) = (I_{n \times n} + \delta^t \cdot \frac{k-1}{n-1} \cdot G(\boldsymbol{s}')) \cdot g(\boldsymbol{s}^t, \boldsymbol{\mu}^t).$$

With $\eta^t = \delta^t \cdot \left(\frac{k-1}{n-1}\right)$, we get $g(\boldsymbol{s}^{t+1}, \boldsymbol{\mu}^t) = (I_{n \times n} + \eta^t \cdot G(\boldsymbol{s}')) \cdot g(\boldsymbol{s}^t, \boldsymbol{\mu}^t)$.

Taking the $L^2$ norm, we get:

$$\|g(s^{t+1}, \boldsymbol{\mu}^t)\|_2^2 = \|g(s^t, \boldsymbol{\mu}^t)\|_2^2 + (\eta^t)^2 \cdot \|G(s')g(s^t, \boldsymbol{\mu}^t)\|_2^2 + 2\eta^t g(s^t, \boldsymbol{\mu}^t)^T G(s')g(s^t, \boldsymbol{\mu}^t), \tag{21}$$

By the strong concavity assumption, for a constant $\lambda > 0$, $G + \lambda \cdot I_{n \times n}$ is negative semi-definite, i.e., $v^T(G + \lambda \cdot I_{n \times n})v \leq 0$ for any $v \in \mathbb{R}^n$. With $v = g(s^t, \boldsymbol{\mu}^t)$, we have:

$$g(s^t, \boldsymbol{\mu}^t)^T G(s')g(s^t, \boldsymbol{\mu}^t) \leq -\lambda \cdot \|g(s^t, \boldsymbol{\mu}^t)\|_2^2. \tag{22}$$

Next we use the fact that the $L^2$ norm $\|A\|_2$ of an $n \times n$ matrix $A$ is bounded by its Frobenius norm $\|A\|_F$:

$$\|A\|_2 := \sup_{x \neq 0} \frac{\|Ax\|_2}{\|x\|_2} \leq \|A\|_F := \sqrt{\sum_i \sum_j |A_{ij}|^2}$$

By the bounded derivatives assumption, we have $|G(s')_{ij}| \leq L$, which implies that $\|G(s')\|_F = \sqrt{\sum_i \sum_j L^2} = nL$. This gives:

$$\|G(s')g(s^t, \boldsymbol{\mu}^t)\|_2 \leq nL\|g(s^t, \boldsymbol{\mu}^t)\|_2. \tag{23}$$

Using (22) and (23) in (21), we get:

$$\|g(s^{t+1}, \boldsymbol{\mu}^t)\|_2^2 \leq (1 + \eta_t^2 \cdot n^2 L^2 - 2\eta^t \lambda) \cdot \|g(s^t, \boldsymbol{\mu}^t)\|_2^2. \tag{24}$$

The definition $\delta^t = \frac{n-1}{k-1} \cdot \frac{\lambda}{n^2 L^2}$ implies that $\eta^t = \frac{\lambda}{n^2 L^2}$. Equation (24) together with Equation (14) gives:

$$\|g(s^{t+1}, \boldsymbol{\mu}^{t+1})\|_2^2 \leq \left(1 - \frac{\lambda^2}{n^2 L^2}\right) \cdot \|g(s^t, \boldsymbol{\mu}^t)\|_2^2.$$

Using the above inequality recursively, and using the inequality $(1 - x)^r \leq e^{-xr}$, we obtain:

$$\|g(s^t, \boldsymbol{\mu}^t)\|_2 \leq e^{-\frac{\lambda^2}{2n^2 L^2} \cdot t} \cdot \|g(s^0, \boldsymbol{\mu}^0)\|_2.$$

Thus in $T = \frac{2n^2 L^2}{\lambda^2} \log \left(\frac{\|g(s^0, \boldsymbol{\mu}^0)\|_2}{\varepsilon}\right)$ iterations, we have $\|g(s^T, \boldsymbol{\mu}^T)\|_2 \leq \varepsilon$, as claimed. $\qquad \square$

**Lemma 4.5.** *For a federated learning instance where agent (i) payoff functions are $\lambda_1$-strongly concave and cost functions are $\lambda_2$-strongly concave, and (ii) second derivatives of payoffs and costs are bounded: $|\frac{\partial^2 a_i}{\partial s_j \partial s_k}| \leq L_1$ and $|\frac{\partial^2 c_i}{\partial^2 s_i}| \leq L_2$, for constants $\lambda_1, \lambda_2, L_1, L_2 > 0$, stochastic best response dynamics (11) with step size $\delta^t = \frac{n-1}{k-1} \cdot \frac{n\lambda_1 + \lambda_2}{n^2(2nL_1 + L_2)^2}$ converges to an approximate Nash equilibrium $s^T$ where $\|g(s^T, \boldsymbol{\mu}^T)\|_2 < \varepsilon$ in $T$ iterations, where:*

$$T = \frac{2n^2 \cdot (2nL_1 + L_2)^2}{(n\lambda_1 + \lambda_2)^2} \log \left(\frac{\|g(s^0, \boldsymbol{\mu}^0)\|_2}{\varepsilon}\right)$$

*Proof.* We first show that each agent's utility function $u_i(s)$ is $(n\lambda_1 + \lambda_2)$-strongly concave as follows: for any $i \in N$,

$$\frac{\partial^2 u_i}{\partial s_i^2} + n\lambda_1 + \lambda_2 = \frac{1}{n} \cdot \sum_{X \subseteq N \setminus \{i\}} \binom{n-1}{|X|}^{-1} \cdot \left(\frac{\partial^2 A(s[X \cup \{i\}])}{\partial s_i^2}\right) - \frac{\partial^2 c_i}{\partial s_i^2} + n\lambda_1 + \lambda_2 \qquad \text{(By Eqn 7)}$$

$$< \frac{1}{n} \cdot \sum_{X \subseteq N \setminus \{i\}} \binom{n-1}{|X|}^{-1} \cdot n(-\lambda_1) - \lambda_2 + n\lambda_1 + \lambda_2 \qquad \text{(By Assumption (i) and (ii))}$$

$$\leq -\lambda_1 \cdot n - \lambda_2 + n \cdot \lambda_1 + \lambda_2 = 0,$$

which concludes the fact. Next, we show that the second derivatives of $u_i$ are bounded. Observe that:

$$\left| \frac{\partial^2 u_i}{\partial s_j \partial s_k} \right| = \left| \frac{\partial^2 \varphi_i^A(\boldsymbol{s})}{\partial s_j \partial s_k} - \frac{\partial^2 c_i}{\partial s_j \partial s_k} \right|$$

$$\leq \frac{1}{n} \cdot \sum_{X \subseteq N \setminus \{i\}} \binom{n-1}{|X|}^{-1} \cdot \left| \left( \frac{\partial^2 A(\boldsymbol{s}[X \cup \{i\}])}{\partial s_j \partial s_k} - \frac{\partial^2 A(\boldsymbol{s}[X])}{\partial s_j \partial s_k} \right) \right| + \left| \frac{\partial^2 c_i}{\partial s_j \partial s_k} \right|$$

$$\leq \frac{1}{n} \cdot \sum_{X \subseteq N \setminus \{i\}} \binom{n-1}{|X|}^{-1} \cdot 2nL_1 + L_2 \qquad \text{(By Assumption (ii) and } A = \sum_i a_i\text{)}$$

$$\leq \frac{1}{n} \cdot n \cdot 2nL_1 + L_2 = 2nL_1 + L_2 \, .$$

With Theorem 4.4 and the above two inequalities, Lemma 4.5 follows. $\qquad\square$

**Lemma 4.6.** $\mathcal{M}^{\mathsf{Shap}}$ *is individually rational.*

*Proof.* Let $\boldsymbol{s}^* \in \mathsf{NE}(\mathcal{M}^{\mathsf{Shap}})$. Since $\boldsymbol{s}^*$ is a NE, each agent $i$ does not benefit from deviating unilaterally. Therefore, $u_i(\boldsymbol{s}^*) \geq u_i(\boldsymbol{s}')$, where $s_i' = 0$ and $s_j' = s_j^*$ for all $j \neq i$. This shows $u_i(\boldsymbol{s}^*) \geq a_i(\boldsymbol{s}') - c_i(s_i') = a_i(\boldsymbol{s}') \geq 0$. Thus, $\mathcal{M}^{\mathsf{Shap}}$ is individually rational. $\qquad\square$

**Theorem 4.7.** $\mathcal{M}^{\mathsf{Shap}}$ *satisfies* $\mathsf{Reciprocity}(\mathcal{M}^{\mathsf{Shap}}) = 1$*, i.e., is fully reciprocal. Moreover,* $\mathcal{M}^{\mathsf{Shap}}$ *satisfies equal treatment of equals.*

*Proof.* Consider any NE $\boldsymbol{s} \in \mathsf{NE}(\mathcal{M}^{\mathsf{Shap}})$. By definition of the payment rule of $\mathcal{M}^{\mathsf{Shap}}$ given by Equation (7), $a_i(\boldsymbol{s}) + p_i(\boldsymbol{s}) = \varphi_i^A(\boldsymbol{s})$ for every $i \in N$. Thus $\mathsf{Reciprocity}(\mathcal{M}^{\mathsf{Shap}}) = 1$.

To see why $\mathcal{M}^{\mathsf{Shap}}$ satisfies equal treatment of equals (Definition 3.5), consider two identical agents $i$ and $j$, i.e., $a_i(\cdot) = a_j(\cdot)$, $c_i(\cdot) = c_j(\cdot)$, and $s_i = s_j$. Then, at a NE $\boldsymbol{s} \in \mathsf{NE}(\mathcal{M}^{\mathsf{Shap}})$, we have:

$$p_i(\boldsymbol{s}) = \varphi_i^A(\boldsymbol{s}) - c_i(s_i) = \frac{1}{n} \cdot \sum_{X \subseteq N \setminus \{i\}} \binom{n-1}{|X|}^{-1} \cdot \Big( A(\boldsymbol{s}[X \cup \{i\}]) - A(\boldsymbol{s}[X]) \Big) - c_i(s_i)$$

$$= \frac{1}{n} \cdot \sum_{X \subseteq N \setminus \{j\}} \binom{n-1}{|X|}^{-1} \cdot \Big( A(\boldsymbol{s}[X \cup \{j\}]) - A(\boldsymbol{s}[X]) \Big) - c_j(s_j)$$

$$= \varphi_j^A(\boldsymbol{s}) - c_j(s_j) = p_j(\boldsymbol{s}),$$

where we replaced $i$ with $j$ in the penultimate step as they are identical agents. $\qquad\square$

**Theorem 4.8.** *Let* $W(\boldsymbol{s}) = A(\boldsymbol{s}) - \sum_{i \in [n]} c_i(s_i)$ *denote the total welfare of the agents, and let* $\boldsymbol{s}^* \in \mathsf{NE}(\mathcal{M}^{\mathsf{Shap}})$. *Consider any data contribution vector* $\boldsymbol{s}$ *that weakly Pareto-dominates* $\boldsymbol{s}^*$*, i.e.,* $s_i \geq s_i^*$ *for all* $i$*. Then* $W(\boldsymbol{s}) < W(\boldsymbol{s}^*)$.

*Proof.* We first prove a useful Lemma.

**Lemma D.1.** *For any sample vector* $\boldsymbol{s} \in \mathcal{S}$ *and agent* $i \in N$*,* $\frac{\partial A(\boldsymbol{s})}{\partial s_i} \leq \frac{\partial \varphi_i^A(\boldsymbol{s})}{\partial s_i}$.

*Proof.* We use the definition of the Shapley value of federation as follows to prove the lemma.

$$\frac{\partial \varphi_i^A(\boldsymbol{s})}{\partial s_i} = \frac{1}{n} \cdot \sum_{X \subseteq N \setminus \{i\}} \binom{n-1}{|X|}^{-1} \cdot \left( \frac{\partial A(\boldsymbol{s}[X \cup \{i\}])}{\partial s_i} - \frac{\partial A(\boldsymbol{s}[X])}{\partial s_i} \right)$$

$$= \frac{1}{n} \cdot \sum_{X \subseteq N \setminus \{i\}} \binom{n-1}{|X|}^{-1} \cdot \frac{\partial A(\boldsymbol{s}[X \cup \{i\}])}{\partial s_i} \qquad \text{(since } i \notin X\text{)}$$

$$\geq \frac{1}{n} \cdot \sum_{X \subseteq N \setminus \{i\}} \binom{n-1}{|X|}^{-1} \cdot \frac{\partial A(\boldsymbol{s}[N])}{\partial s_i} \quad \text{(using concavity of } A)$$

$$= \frac{\partial A(\boldsymbol{s}[N])}{\partial s_i}. \qquad \square$$

Since $W(s)$ is strictly concave, we have:

$$W(s) < W(s^*) + \nabla W(s^*)^T(s - s^*)$$

$$= W(s^*) + \left\langle \left( \frac{\partial(A(s) - c_1(s_1))}{\partial s_1}, \frac{\partial(A(s) - c_2(s_2))}{\partial s_2}, \dots, \frac{\partial(A(s) - c_n(s_n))}{\partial s_n} \right) \Big|_{s=s^*}, s - s^* \right\rangle$$

Since $s_i - s_i^* \geq 0$ for all $i \in [n]$, using Lemma D.1, we have:

$$W(s) < W(s^*) + \left\langle \left( \frac{\partial(\varphi_1^A(s) - c_1(s_1))}{\partial s_1}, \frac{\partial(\varphi_2^A(s) - c_2(s_2))}{\partial s_2}, \dots, \frac{\partial(\varphi_n^A(s) - c_n(s_n))}{\partial s_n} \right) \Big|_{s=s^*}, s - s^* \right\rangle$$

$$= W(s^*)$$

where the last inequality follows from the fact that $s^*$ is an NE. $\qquad \square$

**Theorem 4.9.** *Consider any FL instance with $n$ agents where agents have (i) identical payoff function $a(\boldsymbol{s}) = 1 - \alpha \cdot (\|\boldsymbol{s}\|_1 + 1)^{-\beta}$ for $\alpha > 0$ and $\beta \in (0, 1]$, and (ii) linear cost functions $c_i(s_i) = \gamma_i \cdot s_i + d_i$ for $\gamma_i, d_i \geq 0$. Then $\mathcal{M}^{\mathsf{Shap}}$ satisfies:*

*(i)* $\mathsf{DataGain}(\mathcal{M}^{\mathsf{Shap}}) \geq n^{\frac{1}{\beta+1}}$, *and*

*(ii)* $\mathsf{AccGain}(\mathcal{M}^{\mathsf{Shap}}) \geq 1 + \alpha^{\frac{1}{\beta+1}} \cdot \beta^{\frac{-\beta}{\beta+1}} \cdot (\min_i \gamma_i)^{\frac{\beta}{\beta+1}} \cdot \left(1 - n^{-\frac{\beta}{1+\beta}}\right)$.

*Proof.* We first show part (i). Let $K = \arg\min_{i \in N} \gamma_i$ be the set of agents with the least marginal cost denoted by $\gamma_k = \min_{i \in N} \gamma_i$. Consider a NE $\boldsymbol{s}^0$ of the mechanism $\mathcal{M}^0$ without payments. At the NE, no agent has any incentive to change their contribution, i.e., $\frac{\partial a_i(\boldsymbol{s})}{\partial s_i} = \frac{\partial c_i(s_i)}{\partial s_i}$. Using this condition, we observe that $\frac{\alpha \cdot \beta}{(\|\boldsymbol{s}^0\|_1 + 1)^{\beta+1}} = \gamma_i = \gamma_k$ for all $i \in K$. Moreover, $s_i^0 = 0$ for all $i \notin K$, since an agent $i \notin K$ has no incentive to contribute any data points. Thus, an NE $\boldsymbol{s}^0$ of $\mathcal{M}^0$ satisfies:

$$\|\boldsymbol{s}^0\|_1 + 1 = \left( \frac{\alpha\beta}{\gamma_k} \right)^{\frac{1}{\beta+1}}. \tag{25}$$

Let us now consider any NE $\boldsymbol{s}^*$ of $\mathcal{M}^{\mathsf{Shap}}$. By definition of NE, we have that $\frac{\partial \varphi_i^A(\boldsymbol{s}^*)}{\partial s_i} = \frac{\partial c_i(\boldsymbol{s}^*)}{\partial s_i}$ for all $i \in N$. Using linearity of costs and Lemma D.1, we note that $\frac{\partial A(\boldsymbol{s}^*)}{\partial s_i} \leq \gamma_i$ for all $i$. Explicitly computing the derivative gives us:

$$n \cdot \frac{\alpha \cdot \beta}{(\|\boldsymbol{s}^*\|_1 + 1)^{\beta+1}} \leq \min_i \gamma_i = \gamma_k.$$

In turn, this implies that $\|\boldsymbol{s}^*\|_1 + 1 \geq \left( \frac{n \cdot \alpha \cdot \beta}{\gamma_k} \right)^{\frac{1}{\beta+1}} = n^{\frac{1}{\beta+1}} \cdot \left( \frac{\alpha \cdot \beta}{\gamma_k} \right)^{\frac{1}{\beta+1}} = n^{\frac{1}{\beta+1}} \cdot (\|\boldsymbol{s}^0\|_1 + 1)$, using (25). Since $n \geq 1$, this implies $\|\boldsymbol{s}^*\|_1 \geq \|\boldsymbol{s}^0\|_1$. Therefore:

$$\mathsf{DataGain}(\mathcal{M}^{\mathsf{Shap}}) = \frac{\|\boldsymbol{s}^*\|_1}{\|\boldsymbol{s}^0\|_1} \geq \frac{\|\boldsymbol{s}^*\|_1 + 1}{\|\boldsymbol{s}^0\|_1 + 1} \geq n^{\frac{1}{\beta+1}}, \tag{26}$$

thus proving part (i).

For part (ii), using the fact that agents have identical payoff functions, the accuracy gain is given by:

$$\mathsf{AccGain}(\mathcal{M}^{\mathsf{Shap}}) = \frac{A(\boldsymbol{s}^*)}{A(\boldsymbol{s}^0)} = \frac{a(\boldsymbol{s}^*)}{a(\boldsymbol{s}^0)}$$

$$\begin{aligned}
&= \frac{1 - \alpha \cdot (\|\boldsymbol{s}^*\|_1 + 1)^{-\beta}}{1 - \alpha \cdot (\|\boldsymbol{s}^0\|_1 + 1)^{-\beta}} \\
&= 1 + \alpha \cdot \frac{(\|\boldsymbol{s}^0\|_1 + 1)^{-\beta} - (\|\boldsymbol{s}^*\|_1 + 1)^{-\beta}}{1 - \alpha \cdot (\|\boldsymbol{s}^0\|_1 + 1)^{-\beta}} \\
&\geq 1 + \frac{\alpha}{(\|\boldsymbol{s}^0\|_1 + 1)^{\beta}} \cdot \left(1 - \frac{(\|\boldsymbol{s}^0\|_1 + 1)^{\beta}}{(\|\boldsymbol{s}^*\|_1 + 1)^{\beta}}\right) \\
&\geq 1 + \frac{\alpha}{(\|\boldsymbol{s}^0\|_1 + 1)^{\beta}} \cdot \left(1 - n^{-\frac{\beta}{\beta+1}}\right) && \text{(By Equation (26))} \\
&\geq 1 + \frac{\alpha}{\left(\frac{\alpha\beta}{\gamma_k}\right)^{\frac{\beta}{\beta+1}}} \cdot \left(1 - n^{-\frac{\beta}{\beta+1}}\right) && \text{(by Equation (25))} \\
&\geq 1 + \alpha^{\frac{1}{\beta+1}} \cdot \beta^{\frac{-\beta}{\beta+1}} \cdot \gamma_k^{\frac{\beta}{\beta+1}} \cdot \left(1 - n^{-\frac{\beta}{\beta+1}}\right),
\end{aligned}$$

thus proving the theorem. $\square$

## E. More Details of Best Response Dynamics

We set the number of clients as 30 for the three image-based datasets. Each client a) in MNIST has 175-191 batches of training data and 17-18 batches of testing data; b) in FashionMNIST has 173-192 batches of training data and 17-18 batches of testing data; c) in CIFAR has 27 batches of training data and 6 batches of testing data.

We use a simple CNN network with two $5 \times 5$ convolution layers followed by two fully connected layers with ReLU activation for MNIST/FashionMNIST and the ResNet-18 (He et al., 2016) for CIFAR-10. We use ResMLP (Touvron et al., 2022) for local training of the healthcare dataset and a simple quadratic regression for the synthetic dataset.

**Fitting the accuracy functions.** As described in Section 5.1, we perform a preprocessing training step to fit the accuracy functions in advance. Specifically, we first conduct a lightweight standard FL training without strategic sharing, using 0 and 200 batches from each group, respectively. We run the training for 100 epochs and fit the closed-form accuracy functions using the collected results. For the four non-synthetic datasets, the parameters of the closed-form accuracy functions are as follows:

- MNIST:
$$\begin{bmatrix} w_{11} & w_{12} & w_{13} \\ w_{21} & w_{22} & w_{23} \\ w_{31} & w_{32} & w_{33} \end{bmatrix} = \begin{bmatrix} 3.1 \times 10^{-3} & 4.7 \times 10^{-4} & 4.4 \times 10^{-4} \\ 6.7 \times 10^{-4} & 1.9 \times 10^{-3} & 6.5 \times 10^{-4} \\ 9.3 \times 10^{-4} & 8.3 \times 10^{-4} & 2.1 \times 10^{-3} \end{bmatrix}$$

- FashionMNIST:
$$\begin{bmatrix} w_{11} & w_{12} & w_{13} \\ w_{21} & w_{22} & w_{23} \\ w_{31} & w_{32} & w_{33} \end{bmatrix} = \begin{bmatrix} 1.3 \times 10^{-3} & 3.1 \times 10^{-4} & 5.7 \times 10^{-4} \\ 1.0 \times 10^{-4} & 1.5 \times 10^{-3} & 2.6 \times 10^{-4} \\ 4.9 \times 10^{-4} & 4.4 \times 10^{-4} & 1.6 \times 10^{-3} \end{bmatrix}$$

- CIFAR-10:
$$\begin{bmatrix} w_{11} & w_{12} & w_{13} \\ w_{21} & w_{22} & w_{23} \\ w_{31} & w_{32} & w_{33} \end{bmatrix} = \begin{bmatrix} 5.8 \times 10^{-3} & 1.4 \times 10^{-3} & 7.9 \times 10^{-4} \\ 2.5 \times 10^{-3} & 6.1 \times 10^{-3} & 7.4 \times 10^{-4} \\ 4.2 \times 10^{-3} & 3.0 \times 10^{-3} & 1.8 \times 10^{-3} \end{bmatrix}$$

- Lumpy-Skin-Disease:
$$\begin{bmatrix} w_{11} & w_{12} \\ w_{21} & w_{22} \end{bmatrix} = \begin{bmatrix} 1.2 \times 10^{-2} & 9.1 \times 10^{-3} \\ 1.4 \times 10^{-2} & 1.6 \times 10^{-2} \end{bmatrix}$$

The cost of adding one training data patch $\gamma_i$ is chosen uniformly at random from $[0, 0.001]$ for MNIST, FashionMNIST, and CIFAR-10. We run the best response dynamics for all three mechanisms for 1000 iterations. We set the step size $\delta$ of best response dynamics to be 10, and the learning rate $\alpha$ is set as 0.1 for the local training.

**System configuration.** Our experiments were conducted on the Illinois Campus Cluster configured with one node with 16 cores, Fedora@9.4 operating system, and one A100 GPU.

**Approximation of Shapley value.** We adopt a simple Monte Carlo estimate for the Shapley value by uniformly sampling a set of permutations of clients $\Pi$ (Mann & Shapley, 1960; Maleki, 2015; Jia et al., 2019; Zhang et al., 2023). Let $P_i^\sigma$ be the set of clients located in front of $i$ in the permutation $\sigma$. The approximate Shapley value of client $i$ is given by:

$$\hat{\varphi}_i^A(s) = \frac{1}{|\Pi|} \sum_{\sigma \in \Pi} \left(A(s[P_i^\sigma \cup \{i\}] - A(s[P_i^\sigma])\right) . \tag{27}$$

Theoretically, for $n$ agents, $m = \frac{2n}{\epsilon^2} \ln \frac{2n}{\delta}$ samples ensure an error of $\epsilon$ and confidence of $1 - \delta$. However, in the implementation, we adopt a more ambitious setting by sampling only $\lfloor n \cdot \log n \rfloor = 102$ permutations for the three image-base datasets, as mentioned in Sec 5.1 of the first three image-base datasets. To justify the soundness of the setting, we report the standard deviation in Tables 2 to 4. It can be observed that all the statistics of $\mathcal{M}^{\mathsf{Shap}}$ has a standard deviation of no more than 0.1, which demonstrates the sufficiency of the current number of samples.

| Method | DataShare(%) | | | Accuracy(%) | | | Reciprocity | | | DataGain | | | AccGain | | |
|---|---|---|---|---|---|---|---|---|---|---|---|---|---|---|---|
| | Avg. | Med. | $\sigma$ | Avg. | Med. | $\sigma$ | Avg. | Med. | $\sigma$ | Avg. | Med. | $\sigma$ | Avg. | Med. | $\sigma$ |
| $\mathcal{M}^0$ | 5.8 | 5.6 | 0.002 | 88.3 | 88.5 | 0.004 | 0.609 | 0.630 | 0.041 | 1.000 | 1.000 | 0.000 | 1.000 | 1.000 | 0.000 |
| $\mathcal{M}^{\mathsf{BG}}$ | 7.6 | 7.6 | 0.002 | 87.6 | 87.6 | 0.018 | 0.702 | 0.706 | 0.012 | 1.324 | 1.349 | 0.043 | 0.992 | 0.990 | 0.024 |
| $\mathcal{M}^{\mathsf{Shap}}$ | 54.9 | 54.9 | 0.000 | 90.4 | 90.5 | 0.006 | 1.000 | 1.000 | 0.000 | 9.514 | 9.719 | 0.354 | 1.024 | 1.022 | 0.004 |

*Table 2.* Standard deviation of MNIST

| Method | DataShare(%) | | | Accuracy(%) | | | Reciprocity | | | DataGain | | | AccGain | | |
|---|---|---|---|---|---|---|---|---|---|---|---|---|---|---|---|
| | Avg. | Med. | $\sigma$ | Avg. | Med. | $\sigma$ | Avg. | Med. | $\sigma$ | Avg. | Med. | $\sigma$ | Avg. | Med. | $\sigma$ |
| $\mathcal{M}^0$ | 4.1 | 3.9 | 0.003 | 60.9 | 61.3 | 0.017 | 0.466 | 0.467 | 0.030 | 1.000 | 1.000 | 0.000 | 1.000 | 1.000 | 0.000 |
| $\mathcal{M}^{\mathsf{BG}}$ | 6.2 | 6.2 | 0.001 | 61.5 | 61.1 | 0.011 | 0.752 | 0.759 | 0.020 | 1.513 | 1.558 | 0.078 | 1.023 | 1.023 | 0.012 |
| $\mathcal{M}^{\mathsf{Shap}}$ | 54.8 | 54.8 | 0.000 | 63.3 | 63.4 | 0.016 | 1.000 | 1.000 | 0.000 | 13.436 | 13.895 | 0.795 | 1.054 | 1.069 | 0.055 |

*Table 3.* Standard deviation of FashionMNIST

| Method | DataShare(%) | | | Accuracy(%) | | | Reciprocity | | | DataGain | | | AccGain | | |
|---|---|---|---|---|---|---|---|---|---|---|---|---|---|---|---|
| | Avg. | Med. | $\sigma$ | Avg. | Med. | $\sigma$ | Avg. | Med. | $\sigma$ | Avg. | Med. | $\sigma$ | Avg. | Med. | $\sigma$ |
| $\mathcal{M}^0$ | 25.6 | 25.7 | 0.005 | 43.6 | 43.9 | 0.009 | 0.504 | 0.505 | 0.007 | 1.000 | 1.000 | 0.000 | 1.000 | 1.000 | 0.000 |
| $\mathcal{M}^{\mathsf{BG}}$ | 28.9 | 28.9 | 0.001 | 44.7 | 44.6 | 0.003 | 0.566 | 0.569 | 0.016 | 1.131 | 1.125 | 0.018 | 1.025 | 1.023 | 0.017 |
| $\mathcal{M}^{\mathsf{Shap}}$ | 99.6 | 99.6 | 0.001 | 48.5 | 48.7 | 0.011 | 1.000 | 1.000 | 0.000 | 3.894 | 3.870 | 0.078 | 1.114 | 1.112 | 0.007 |

*Table 4.* Standard deviation of CIFAR-10

# F. `FedBR-Shap` Protocol

## F.1. Motivation of `FedBR-Shap`

Still, in this section, we interpret $s_i$ as the number of batches of the local training of client $i$. As discussed before, the computation of NE in Section 5.1 relies on static closed-form accuracy functions of $s$ and forces every client to use a fixed $s_i$ throughout the training. However, the influence of data sharing on the accuracy can differ across stages: data sharing can be quite beneficial in the initial phases, but may become less effective as the model converges. As a result, a data client may also exhibit strategic behaviors at different stages of the training, e.g., reducing the number of batches of local training when the model turns to converge. In addition, the aforementioned mechanisms only make payments at the end of the training, which makes it hard to evaluate a client's contribution to the training over various stages of the training.

Motivated by the above aspects, we propose the protocol `FedBR-Shap`, which is a truly distributed FL protocol. `FedBR-Shap` allows clients to adjust the value of $s_i$ and also provide (budget-feasible) payments in different stages

of the training. The payment to agent $i$ at iteration $t$ depends on the contribution of agent $i$'s data sharing towards the accuracy improvement at that iteration. We make the following assumption for the learning rate:

**Assumption F.1.** Assume the learning rate of model updating is set as small enough such that, within any window of $W$ iterations, the improvement of the accuracy between two consecutive iterations, $a_i(\theta^t) - a_i(\theta^{t-1})$ can be approximated by some identical function $\alpha_i(s^t)$ that only depends on the sample vector at iteration $t$.

### F.2. Description of `FedBR-Shap`

`FedBR-Shap` computes the NEs of $\mathcal{M}^{\mathsf{Shap}}$ following best response dynamics (Sec 4.2). In each iteration $t$, the algorithm computes a global model $\theta^t$, and maintains copies of local models $T_i$ trained only on agent $i$'s dataset of size $s_i^t$, starting from the global model from the previous iteration, $\theta^{t-1}$. The central server then updates its global model, and the agents update their data shares according to their current gradients of the utility functions. As motivated in Appendix F.1, the growth of accuracy varies during the training process. For this reason, we divide the entire training into a series of stages and decide the payments at the end of each stage. Below, we first define our payment mechanism, which essentially adopts the same idea of $\mathcal{M}^{\mathsf{Shap}}$. Thereafter, we provide the implementation of `FedBR-Shap` using best response dynamics.

**Payments.** A *stage* is defined as a sequence of $W$ iterations. The central server distributes payments at every stage according to everyone's contribution to accuracy improvement at that stage. Denote the accuracy of a model $\theta$ for agent $i$ by $a_i(\theta)$ and the model after the $t$-th iteration by $\theta^t$. The total accuracy increase at stage $h$ is given by $\sum_{i=1}^n a_i(\theta^t) - \sum_{i=1}^n a_i(\theta^{t-W})$, where $t = W \cdot h$. The central server then gives the payment

$$p_i^h = \varphi_i^h(s_i^t, s_{-i}^t) - (a_i(\theta^t) - a_i(\theta^{t-W}))$$

to agent $i$, where $\varphi_i^h(s_i^t, s_{-i}^t)$ denotes the Shapley value of agent $i$ towards the total accuracy increase at stage $h$, which is given by $\sum_{i \in [n]} a_i(\theta^t) - a_i(\theta^{t-W})$.

**Best response dynamics.** We follow the best response dynamics within every stage to compute an NE of agents' strategies for the current training stage. If the clients take sample vector $s$, by Assumption F.1, the total accuracy increase $\sum_{i=1}^n a_i(\theta^t) - a_i(\theta^{t-W})$ is given by $W \cdot \sum_{i=1}^n \alpha_i(s)$. Denote by $\boldsymbol{\alpha}(\cdot)$ the sum of $\alpha_i(\cdot)$. Then the utility of client $i$ is

$$u_i^h(s) = W \cdot \varphi_i^{\boldsymbol{\alpha}}(s_i, s_{-i}) - \gamma_i \cdot s_i.$$

Based on the utility function, `FedBR-Shap` performs $W$ rounds of best response dynamics to update $s$. Each agent $i$ updates to $s_i^t$ by computing the gradient of the current utility function, $\nabla_i u_i^h(s^t)$, and hence the gradient of its Shapley share $\nabla_i \varphi_i^{\boldsymbol{\alpha}}(s^t)$. This is estimated by the server as a difference of Shapley shares, as

$$\nabla_i \varphi_i^{\boldsymbol{\alpha}}(s^t) \approx \frac{1}{\varepsilon} \cdot \left( \varphi_i^{\boldsymbol{\alpha}}(s_i^t + \varepsilon, s_{-i}^t) - \varphi_i^{\boldsymbol{\alpha}}(s_i^t, s_{-i}^t) \right).$$

**Approximation of Shapley share.** Now consider an agent $i$, for which the goal is to compute the $\nabla_i \varphi_i^{\boldsymbol{\alpha}}(s^t)$. By Assumption F.1, we have $\boldsymbol{\alpha}(s) \approx \sum_{j=1}^n a_j(\theta^{t+1}) - a_j(\theta^t)$. Denote by $\theta_X^t$ the model after aggregating the local models of clients from a set $X$ starting from the global model $\theta^t$. Therefore, the Shapley value of stage $h$ can be approximated as follows:

$$\varphi_i^{\boldsymbol{\alpha}}(s_i^t, s_{-i}^t) \approx \sum_{j=1}^n \varphi_i^{a_j(\theta^{t+1}) - a_j(\theta^t)}(s_i^t, s_{-i}^t)$$

$$= \frac{1}{2^{n-1}} \sum_{X \subseteq [n] \setminus \{i\}} \sum_{j=1}^n \left( a_j(\theta_{X \cup \{i\}}^t) - a_j(\theta_X^t) \right)$$

To compute the derivative, beginning with the current global model $\theta^t$, each agent $i \in [n]$ updates its local model $T_i$ using $s_i^t$ batches of its dataset. Moreover, each agent $i$ trains an extra model of $T_i^\varepsilon$ on $s_i^t + \varepsilon$ batches of its local dataset, starting with $\theta^{t-1}$. All these models are transmitted to the server. For each subset $X \subseteq [n]$ where $i \notin X$, the server computes a single model: $\theta_X^t = \frac{1}{|X|} \sum_{j \in X} T_j$ by averaging the model parameters from the agents in $X$. For each subset $X \subseteq [n]$ where $i \in X$, the server computes two models: $\theta_X^t = \frac{1}{|X|} \sum_{j \in X} T_j$, and $\hat{\theta}_X^t = \frac{1}{|X|}(T_i^\varepsilon + \sum_{j \in X \setminus \{i\}} T_j)$. The server then distributes these models to all agents, who report back with their accuracies.

With this information, the server computes $\varphi_i^{\boldsymbol{\alpha}}(\boldsymbol{s}^t)$ and $\varphi_i^{\boldsymbol{\alpha}}(s_i^t + \epsilon, s_{-i}^t)$ using Eq. (3). Note that for a subset $X \subseteq [n]$ where $i \in X$, the cumulative accuracy from $\theta_X^t$ is used for computing $\varphi_i^{\boldsymbol{\alpha}}(s_i^t, s_{-i}^t)$, while the cumulative accuracy from $\hat{\theta}_X^t$ is used for computing $\varphi_i^{\boldsymbol{\alpha}}(s_i^t + \varepsilon, s_{-i}^t)$. For each subset $X \subseteq [n]$ where $i \notin X$, the cumulative accuracy of $\theta_X^t$ is used in the computation of both $\varphi_i^{\boldsymbol{\alpha}}(s_i^t, s_{-i}^t)$ and $\varphi_i^{\boldsymbol{\alpha}}(s_i^t + \varepsilon, s_{-i}^t)$. Since agents are aware of their costs, each agent computes $\nabla_i u_i(\boldsymbol{s}^t)$ and updates its data share to $s_i^{t+1}$ using (11).

---

**Algorithm 1** `FedBR-Shap` protocol

1: **Input:** number of iterations $N$, observation size $W$, learning rate $\alpha$, step size $\delta$, $\varepsilon \in (0, 1)$, $n$ agents;
2: **Output:** Model weights $\theta^t$ and individual contributions $\{\boldsymbol{s}^h\}_{h=1}^H$;
3: $\boldsymbol{s}^1 \leftarrow (1, \ldots, 1)$, initialize $\theta^0$ as a zero-model and set $t \leftarrow 1$;
4: Each agent $i \in [n]$ transmits its local model parameters $T_i$ to the server after training on $s_i^1$ batches of data, initialized from the current global model $\theta^0$;
5: $H \leftarrow \lceil N/W \rceil$;
6: **while** $h \leq H$ **do**
7:    ▷ *Best response dynamics*
8:    **for** $t = W \cdot (h-1)$ to $W \cdot H$ **do**
9:       Each agent $i \in [n]$ transmits its local model parameters $T_i$ to the server after training on $s_i^t$ batches of data, initialized from the current global model $\theta^{t-1}$ along with model parameters $T_i^\epsilon$ to the server, trained on $\epsilon$ more batches of data;
10:       The central server updates the global model as $\theta^t \leftarrow \sum_{i \in R^t} T_i / n$;
11:       **for** $i \in [n]$ **do**
12:          The central server computes $\frac{\partial \varphi_i^h(s_i^t, \boldsymbol{s}_{-i}^t)}{\partial s_i} = \frac{\varphi_i^h(s_i^t + \varepsilon, \boldsymbol{s}_{-i}^t) - \varphi_i^h(s_i^t, \boldsymbol{s}_{-i}^t)}{\varepsilon}$ as described in Section 5.2, and sends it along the new global model $\theta^t$ to agent $i$;
13:          Agent $i$ computes $\frac{\partial u_i}{\partial s_i} \leftarrow \frac{\partial \varphi_i^h(s_i^t, \boldsymbol{s}_{-i}^t)}{\partial s_i} - \frac{\partial c_i}{\partial s_i}$;
14:          **if** $s_i^t + \delta \cdot \frac{\partial u_i}{\partial s_i} < 0$ **then**
15:             $s_i^{t+1} \leftarrow 0$;
16:          **else if** $s_i^t + \delta \cdot \frac{\partial u_i}{\partial s_i} > \tau_i$ **then**
17:             $s_i^{t+1} \leftarrow \tau_i$;
18:          **else**
19:             $s_i^{t+1} \leftarrow s_i^t + \delta \cdot \frac{\partial u_i}{\partial s_i}$;
20:          **end if**
21:          $t \leftarrow t + 1$;
22:       **end for**
23:    **end for**
24:    ▷ *Payment according to the contribution of accuracy improvement*
25:    Central server computes the Shapley share of each agent $i$, $\varphi_i^h(s_i^t, s_{-i}^t)$;
26:    Each agent $i$ is paid $p_i^h = \varphi_i^h(s_i^t, s_{-i}^t) - (a_i(\theta^t) - a_i(\theta^{t-W}))$;
27: **end while**
28: **return** the model weights $\theta^t$ and $\{\boldsymbol{s}^h\}_{h=1}^H$;

---

