# OpenReview forum: "You Get What You Give: Reciprocally Fair Federated Learning"
_ICML.cc/2025/Conference — ICML 2025 poster_

### Official Review · Reviewer_ir71 · 2025-03-09

**Overall Recommendation:** 4

**Summary:**

The paper tackles the free-rider problem in a multi-agent federated learning environment by introducing MShap, a Shapley value-based, budget-balanced payment mechanism to enhance fairness and data gains. This mechanism achieves a Nash Equilibrium without requiring knowledge of agents' private cost functions. Experiments were conducted using the MNIST, FashionMNIST, and CIFAR-10 datasets.

## update after rebuttal:
I read the rebuttal of the authors as well as the questions of the reviewers. I maintain my score.

**Claims And Evidence:**

The main claims of the paper are:

(1) MShap gives reciprocal fairness by rewarding agents in proportion to their contribution.
     The payment mechanism is designed to align each agent’s utility with their Shapley value, which guarantees fairness by construction.
     The paper provides theoretical proofs and experimental results that confirm this claim.

(2) MShap achieves Nash equilibria
    The paper proves it using Kakutani’s fixed-point theorem.

**Essential References Not Discussed:**

N/A

**Experimental Designs Or Analyses:**

The paper uses 3 datasets for experiments. It designs distributed protocol (FedBR-Shap) that "relies exclusively on gradient information, eliminating the need for sharing actual data points". Comparisons to baselines such as FL without payments and a welfare-maximizing mechanism have been done.

**Methods And Evaluation Criteria:**

The paper uses Shapley value to achieve fairness, game theory for Nash equilibrium, and best-response dynamics for computation.
The datasets MNIST, FashionMNIST, CIFAR-10 have been used to evaluate the claims of the paper.

**Other Comments Or Suggestions:**

N/A

**Other Strengths And Weaknesses:**

Strengths: The paper is written well and combines existing ideas in game theory and shapley values to address fairness in federated learning.

Some weaknesses are lack of real-world validations, e.g., on healthcare etc.
Shapley value estimation can be computationally expensive.
There is no discussion on scalability when thousands of agents are involved

**Questions For Authors:**

(1) The paper makes the claim that no other mechanism can simultaneously Pareto-dominate MShap in both data share and total welfare.
However, only two baselines have been used for comparison. More complex baselines are missing.

(2) Datasets on real-world use cases, e.e., healthcare, could show how the results could generalize to new situations

(3) A formal complexity analysis of the Shapley value computation in FL settings is missing

**Relation To Broader Scientific Literature:**

The paper introduces reciprocal fairness by building on prior work on mechanism design such as Karimireddy et al., 2022; Murhekar et al., 2023. The use of Shapley values aligns with prior work on agent's contribution and  data valuation such as Wang et al.

**Theoretical Claims:**

For example, theorem 4.3 on the existence of the Nash equilibrium. The proof makes use of Kakutani’s fixed-point theorem.

---

> ### Author Rebuttal · Authors · 2025-04-01
>
> Thank you for your time and comments. We respond to your questions below.
>
> **Q:** The paper makes the claim that no other mechanism can simultaneously Pareto-dominate MShap in both data share and total welfare. However, only two baselines have been used for comparison. More complex baselines are missing.
>
> **A:**
> We would like to point out that our claim of "no other mechanism can simultaneously Pareto-dominate $\mathcal{M}^{Shap}$ in both data share and total welfare" is formally proven in Theorem 4.8. Since the claim is theoretically proven, it holds for *all* comparative baselines. Nevertheless, to empirically demonstrate the guarantees of our mechanism, we have included comparisons to two existing mechanisms for federated learning. We also conducted many new experiments, for which we request the reviewer to kindly refer to the response to reviewer Dvs1.
>
> **Q:** Datasets on real-world use cases, e.e., healthcare, could show how the results could generalize to new situations
>
> **A:** Thank you for the suggestion. As per your suggestion, we added one benchmark on lumpy skin disease prediction [4], where the input is the information of patients (regions, detection metrics, etc), and the output is whether to be identified as lumpy skin disease. We observe that there are unbalanced labels in the training datasets (much more negative data than positive data). Intuitively, positive data points are more helpful for forecasting disease, and our experiment also verifies this point.
>
> We have two agents, each with 2000 data points. 70% of agent 1's data is positive, and 30% of agent 2's is positive. We train the model with varying numbers of samples from the two agents and learn closed-form accuracy functions given by:
> $$
> a_1(\textbf{s}) = 1- \frac{1}{0.0129s_1 + 0.0050s_2 + 1},
> a_2(\textbf{s}) = 1- \frac{1}{0.0148s_1 + 0.0146s_2 + 1}.
> $$
> Observe that the weight of $s_1$ is larger than that of $s_2$, which indicates agent 1's dataset is more valuable than agent 2. We ran best response dynamics for the three mechanisms and found the following results.
>
> * The results of fraction shares are: (i) FedBR: (30.5%, 18.1%);  (ii) FedBR-BG: (30.6%, 18.1%); (iii) FedBR-Shap: (38.1%, 55.4%).
> * The accuracies of the two agents are: (i) FedBR: (93.5%, 95%); (ii) FedBR-BG: (93.5\%, 93.5\%); (iii) FedBR-Shap: (96\%, 96.25\%).
>
> Thus, our mechanism outperforms the two baselines in terms of both data sharing and accuracy. Importantly, Agent 1 has a smaller fraction than Agent 2 in the equilibrium of our mechanism, which contrasts with the other methods. This aligns with the fairness guarantee of our mechanism: An agent with low-quality data should contribute more data points to balance the benefit (s)he receives from other agents.
>
> The above results, along with the results of many new experiments, are detailed in **[Experiments](https://www.dropbox.com/scl/fi/7q6m19rkdml94zo3zwrl8/submission8749_rebuttal.pdf?rlkey=797frtrobzqx62qdblefp08oy&e=1&st=9qc3dzov&dl=0)**.
>
> **Q:** A formal complexity analysis of the Shapley value computation in FL settings is missing
>
> **A:** Thank you for this question. We note that in the worst case, computing the Shapley value can take exponential time in the number of clients. For this reason, we turn to Monte Carlo estimation for computing the Shapley value approximately, as is common [1,2,3]. As shown in these works, to compute the Shapley value for $n$ agents within an error of $\varepsilon\in(0,1)$ and a confidence of $1-\delta$, it suffices to use $m = \frac{2n}{\varepsilon^2}\cdot\log(\frac{2n}{\delta})$. We will mention this formal analysis in our paper. We use $\varepsilon = \delta = 0.1$ in our new experiments.
>
> **References:**
>
> [1] Addressing The Computational Issues of the Shapley Value With Applications in The Smart Grid, Sasan Maleki. 2015
>
> [2] Towards Efficient Data Valuation Based on the Shapley Value, Jia et al. 2019
>
> [3] Efficient Sampling Approaches to Shapley Value Approximation, Zhang et al. 2023
>
> [4] Lumpy Skin Disease Dataset, Afshari Safavi, Ehsanallah. 2021

---

### Official Review · Reviewer_KV1s · 2025-03-13

**Overall Recommendation:** 4

**Summary:**

This paper proposes a payment-based mechanism for improving fairness in federated learning systems, allow Nash Equilibria that are fairer and incentivize strategic participants to share more data. Their approach is designed to ensure reciprocity, i.e. each agent receives exactly as much utility as their fair Shapley contribution towards the federation. The authors provide many useful theorems and properties, including the existence of efficient Nash Equilibrium, budget balancing and best-response convergence. Finally, they perform empirical evaluations with some popular image datasets and show strong performance compared to two reasonable baselines.

Post-rebuttal Updates:
The authors addressed my questions in a satisfactory manner. I will keep my score and recommend this paper be accepted.

**Claims And Evidence:**

The paper supports all claims made, and provides intuition as well as proofs for them.

**Essential References Not Discussed:**

The paper positions itself very well in related work.

**Experimental Designs Or Analyses:**

I checked the experimental design and results, and they appear sound. The evaluation compares against two reasonable baselines, and the results show the strong performance of the proposed approach on all tasks. The evaluation metrics also convey useful information in a condensed form, which helps in better evaluating the contribution.

**Methods And Evaluation Criteria:**

The proposed method is a natural construction to balance reciprocity and enjoys strong properties. The empirical evaluation is also a useful addition to the mostly theoretical paper.

**Other Comments Or Suggestions:**

Page 2 "in metrics of reciprocity, data gain, accuracy gain, data gain, and" data gain is repeated

In the introduction, reciprocity is denoted with $\beta$, but later on, $r$ is used (Section 3.1)

**Other Strengths And Weaknesses:**

I understand the benefits of not relying on the individual cost functions, however, I don't fully understand why not considering costs is a feature. To me, it seems like a simplification of the reciprocity requirement. The cost is part of the total utility an agent derives, so a perfectly balanced system could also balance this cost out, while ensuring no misreporting. I understand this is hard to do, but why would this not be better if it is possible?

**Questions For Authors:**

1. There could be large costs (in terms of payments due) for someone participating with less data. Since the payment is decided only after training the model, is there any bound on how large this payment might be when agent i commits data s_i? Agents might have limited budgets which may affect their participation in the system.
2. On a related note, how is the payment system communicated to the agents? Are agents able to estimate their expected payments locally? It would be useful to walk through an example of how agents compute their best response share s_i, since this would depend on the payment they expect.

**Relation To Broader Scientific Literature:**

The paper proposes a useful payments-based mechanism for improving fairness in federated learning. This has many applications in real federated learning systems, and could encourage participation by stakeholders (particularly due to the guarantee of individual rationality), while ensuring there are no free-riders. The results show strong adherence to the reciprocity metric, which is evident by construction. I see it having applications in areas where acquisition costs are more homogenous, as that will prevent the feeling of unfairness because of varying input cost for the same return.

**Theoretical Claims:**

Yes, I checked the theoretical proofs for budget balanced mechanisms and reciprocity, and for the admittance of Nash Equilibria. I also tried to check the proofs for the convergence of BR dynamics, and these seem correct to the best of my understanding.

---

> ### Author Rebuttal · Authors · 2025-04-01
>
> Thank you for your time and comments. We respond to your questions below.
>
> **Q:** I understand the benefits of not relying on the individual cost functions, however, I don't fully understand why not considering costs is a feature. To me, it seems like a simplification of the reciprocity requirement. The cost is part of the total utility an agent derives, so a perfectly balanced system could also balance this cost out, while ensuring no misreporting. I understand this is hard to do, but why would this not be better if it is possible?
>
> **A:** We agree that a system that ensures costs are not misreported would be ideal. However, we consider that not relying on costs is a feature, because a fair mechanism under our definition of fairness (i) avoids the burden of verifying costs, (ii) retains its guarantees even if agents misreport costs, (iii) does not penalize high quality, low cost agents while still incentivizing high quality, high cost agents. In contrast, the above features will not hold for a "fair" mechanism where fairness is defined to incorporate costs.
>
> Nevertheless, we agree that fairness can be defined in various ways, and alternative definitions of reciprocity could include agents costs. One such definition could be the minimum ratio of an agent’s total utility to their contribution to the federation’s welfare:
>
> $
> \min_{s \in \mathit{NE}(\mathcal M)} \min_{i \in N} \frac{u_i(s)}{\varphi^W_i(s)} = \frac{a_i(s) + p_i(s) - c_i(s_i)}{\varphi^A_i(s) - c_i(s_i)}
> $
>
> Under this definition, our mechanism $\mathcal{M}^{Shap}$ still achieves reciprocity 1. Further, note that $\min_{s \in \mathit{NE}(\mathcal M)} \min_{i \in N} \frac{a_i(s) + p_i(s)}{\varphi^A_i(s)} \leq 1$ for any weakly-budget balanced mechanism $\mathcal{M}$. Thus, we have:
> \begin{align*}
>     \min_{s \in \mathit{NE}(\mathcal M)} \min_{i \in N}  \frac{a_i(s) + p_i(s) - c_i(s_i)}{ \varphi^A_i(s) - c_i(s_i)} &\leq \min_{s \in \mathit{NE}(\mathcal M)} \min_{i \in N} \frac{a_i(s) + p_i(s)}{\varphi^A_i(s)},
> \end{align*}
>
> This implies that other baseline mechanisms will have even lower reciprocity under the new definition than under the current definition of reprocity (as reported in Table 1). Moreover, all our guarantees on data, welfare, and accuracy will continue to hold under this definition.
>
> **Q:** There could be large costs (in terms of payments due) for someone participating with less data. Since the payment is decided only after training the model, is there any bound on how large this payment might be when agent i commits data s_i? Agents might have limited budgets which may affect their participation in the system.
>
> **A:** At a sample vector $s$, the payment to agent $i$ is given by $p_i = \varphi_i^A(s) - a_i(s)$. If agent $i$ has to pay, i.e., $p_i < 0$, we obtain that an upper bound on $p_i$ is $a_i(s)$. That is, in the worst case, an agent will essentially have to pay $a_i(s)$ to obtain the data from other agents. That said, we agree that studying this problem when agents have limited budgets is a very interesting question for future work.
>
> **Q:** On a related note, how is the payment system communicated to the agents? Are agents able to estimate their expected payments locally? It would be useful to walk through an example of how agents compute their best response share s_i, since this would depend on the payment they expect.
>
> **A:** The payment scheme is published upfront. The share of agents are updated during best response dynamics by communication with the server, i.e., through evaluations on models to compute the Shapley share. However, if the accuracy functions of the agents are common knowledge and have a common test set (e.g. when small number of organizations with standard classification tasks), then agents can estimate their payments locally.

---

### Official Review · Reviewer_KBA3 · 2025-03-15

**Overall Recommendation:** 3

**Summary:**

Summary:

The paper studies a method for incentivizing data contributions in
collaborative/federated learning, while also satisfying fairness criteria.
The authors propose evaluating the contributions from each agent via the Shapley value,
based on the value agents derive from the data, and design a payment scheme where agents
get paid or need to pay based on how much value they are deriving from the aggregate data.
The authors show that a stochastic version of best response dynamics (BRD) converges to a
Nash equilibrium (NE), and provide guarantees on the efficiency of any NE.
They supplement this with experiments on three datasets.


Incentives in FL is an important and timely problem at the intersection of machine
learning and game theory.  However, the novelty of the work is unclear, as several key ideas appear to be adapted from previous work (e.g., Murhekar et al.) without clear explanation of what is fundamentally new. The practicality of using payments in a federated learning scheme is not well-motivated. Moreover, several aspects of the paper, including the definition of reciprocity, the baselines for efficiency, and the need for a stochastic BRD are not properly motivated. For these reasons, I cannot recommend the paper for acceptance.


Detailed comments:

1. The paper would benefit from a more thorough discussion of when payments would be practical in a federated learning scheme. Are there real-world scenarios where payments have been successfully implemented in FL?
- This point is particularly important since some agents may need to pay in addition to contributing data. The authors should provide a compelling use case to support the practicality of such a scheme.
- While I acknowledge that some prior works (e.g., Murhekar et al.) have proposed similar frameworks, this does not eliminate the need for the authors to clearly justify their setting.

2. The novelty of the work is not entirely clear. Several ideas seem to be inspired by Murhekar et al. (e.g., budget-balanced payments, the BRD scheme), and it is unclear which aspects are fundamentally new in this paper.
- It appears that the primary contribution is adapting the techniques from Murhekar et al. to also account for fairness through the Shapley value. Is this understanding correct?
- The concept of rewarding agents based on their contribution is not new in collaborative learning and has been previously explored (see [1, 2, 3] below). Moreover, prior works have also employed the Shapley value in collaborative learning schemes to determine agent rewards.
- It would be helpful if the authors clearly summarized the key novelties and differences in the proof techniques early on in the paper under the "Our Contributions" section.

3. The reciprocity term in Definition 3.3 does not appear to include the costs. Excluding costs from this fairness criterion seems arbitrary, especially since costs are being considered when defining the utility. The authors should clarify the rationale behind this choice.

4. Regarding the BRD scheme, it is not clear what the set R_t represents in stochastic BRD. Why is a stochastic version of BRD necessary instead of using regular BRD?

5. In Definitions 3.6 and 3.7, considering NE in a mechanism without payments as a baseline is somewhat unconventional. Generally, a baseline or comparator is expected to be a stronger benchmark, where the objective is for the algorithm to approximate or approach the baseline's performance.
- Here, the baseline appears weaker since it does not allow payments, which explains why the ratio between performance is greater than 1 in Theorem 4.9.
- A more reasonable baseline might be one that maximizes data/accuracy gain when agents are not strategic.
- Furthermore, Theorem 4.9 is difficult to interpret, and additional discussion on how far the proposed method is from optimality would be beneficial.

6. It would strengthen the paper to provide a more comprehensive discussion of related work, particularly including the following references:
   - [1] Cai et al., 2015, "Optimum statistical estimation with strategic data sources"
   - [2] Chen et al., 2020, "Truthful Data Acquisition via Peer Prediction"
   - [3] Chen et al., 2023, "Mechanism Design for Collaborative Normal Mean Estimation"

**Claims And Evidence:**

See above.

**Essential References Not Discussed:**

See above.

**Experimental Designs Or Analyses:**

See above.

**Methods And Evaluation Criteria:**

See above.

**Other Comments Or Suggestions:**

See above.

**Other Strengths And Weaknesses:**

See above.

**Questions For Authors:**

See above.

**Relation To Broader Scientific Literature:**

See above.

**Theoretical Claims:**

See above.

---

> ### Author Rebuttal · Authors · 2025-04-01
>
> Thank you for your time and comments. We respond to the questions below.
>
> **1.** Indeed, there are blockchain-based mechanisms for FL that involve payments based on contributions, such as FedToken [1] and FedCoin [2]. FedCoin uses a "proof of Shapley" protocol, while FedToken distributes tokens based on performance. Both require an initial budget, unlike our budget-balanced approach, which penalizes poor data quality and rewards high-quality data. Also in IoT, BOppCL [3] incentivizes vehicles in intelligent transportation systems, rewarding those with more useful data via cryptocurrency.
>
> **2.1.** Although we work on the same data-sharing framework as of [4, 5] our specific problem objective and solution concepts are fundamentally different. Our primary contribution is a fair mechanism for FL that admits Nash equilibria with strong guarantees on data shared, accuracy, and welfare.
>
> - Objective: [5] focuses on welfare maximizing mechanisms, while not worrying about fairness or data gain. Our goal is to design fair mechanisms, first and for most, that simultaneously has strong guarantees on data sharing, accuracy, and welfare (Thm 4.8 and Thm 4.9).
> - Solution: [5] only use costs to determine payments, whereas we use contributions towards model accuracy. Thus the two mechanisms are fundamentally different.
> - Techniques: The proofs of guarantees of our mechanism are novel. The proof of convergence of *stochastic* BRD generalizes the proof of [5] on BRD. Lastly, we use the standard (and one of the only) proof technique of using the Kakutani fixed point theorem to prove the existence of Nash equilibrium. The same technique is used by several other works, e.g. [4,5].
>
> **2.2.** We request that you kindly revisit Sec 1.1 on Page 2 where we clarify this point: "We remark that rewarding agents according to their contribution levels has been well motivated and studied in FL (Wang et al., 2019; Sim et al., 2020; Zhang et al., 2020; Yu et al., 2020). However, the crucial difference is that our focus is to design a mechanism that incentivizes *strategic agents*, i.e., agents who strategize their data contributions based on the rewards they get from the federation so that desirable fairness and welfare guarantees are achieved at NE."
>
> **3.** We do not include costs in the definition of reciprocal fairness, because a fair mechanism under our definition of fairness (i) avoids the burden of verifying costs, (ii) retains its guarantees even if agents misreport costs, (iii) does not penalize high quality, low cost agents while still incentivizing high quality, high cost agents, unlike [5]. In contrast, the above features will not hold for a "fair" mechanism where fairness includes costs. We also request you to read our response to Reviewer KV1s for a related discussion.
>
> **4.** The set $R^t$ represents a random subset of $k$ agents chosen in round $t$. In round $t$, we perform BR dynamics (i.e., update the data shares) only for agents in $R^t$. A stochastic version is not necessary in general, but is practically useful in situations with large number of clients since it reduces computational and communication overheads in each round. Note that the convergence of stochastic BRD (Theorem 4.4) also implies the convergence of regular BRD, by setting $k=n$.
>
> **5.** We agree that calling a no-payment mechanism a "baseline" is misleading and will avoid this term. Our goal was to quantify the data and accuracy gains enabled by our incentive mechanism with payments as compared to the mechanism without payments.
>
> Also note that a mechanism maximizing total data or accuracy would require all agents to share their data fully. But with high costs, no budget-balanced mechanism can acheive this without large payments to the agents, which is not budget-balanced. Thus, a more meaningful baseline considers both accuracy and cost, like welfare maximization. In this context, we do provide theoretical guarantees against the welfare-maximizing baseline. Specifically, in Thm 4.8, we prove that a welfare-maximizing mechanism cannot achieve Pareto-dominating data contributions from individual agents compared to our mechanism. In fact, this is one of the unique and compelling features of our mechanism.
>
> **6.** Thank you for highlighting the references. Although their primary focus is not on federated learning, we will include a discussion on aspects related to our work such as incentive mechanisms and avoiding free-riding.
>
> **References:**
>
> [1] FedToken: Tokenized Incentives for Data Contribution in Federated Learning, Pandey et al. 2022
>
> [2] FedCoin: A Peer-to-Peer Payment System for Federated Learning, Liu et al. 2020
>
> [3] BOppCL: Blockchain-Enabled Opportunistic Federated Learning in Intelligent Transportation Systems, Li et al. 2023
>
> [4] Mechanisms that Incentivize Data Sharing in Federated Learning, Karimireddy et al. 2022
>
> [5] Incentives in Federated Learning: Equilibria, Dynamics, and Mechanisms for Welfare Maximization, Murhekar et al. 2023

---

### Official Review · Reviewer_Dvs1 · 2025-03-17

**Overall Recommendation:** 4

**Summary:**

In this paper, $\mathcal{M}^{\text{Shap}}$ is proposed, a budget-balanced payoff mechanism for federated learning scheme: the _data-sharing game_ among strategic agents. The authors theoretically designed and elaborated that their proposed mechanism ensures _reciprocal fairness_: each agent's payoff is equal to its Shapley value contribution, admitting efficient Nash equilibria as well as achieving strong performance guarantees. Empirical results show superior fairness and efficiency compared to baselines on vision benchmark datasets.

## Update after rebuttal
---
I updated the score to **accept** because the authors made a great effort to address all of my concerns.
I have summarized below how my evaluation changes after the rebuttal:
* The proposed method is theoretically sound and intriguing, but the empirical validation was not satisfactory.
* During the rebuttal period, the empirical weaknesses are clearly resolved: the authors have faithfully performed additional experiments (**twice**) and the results are acceptable.
* In addition, the authors have also agreed to narrow the scope of their paper based on the point raised by the review. I hope that this reconciliation will increase the visibility of the contributions of the proposed method in the field.

**Claims And Evidence:**

### Main claim
---
$\mathcal{M}^{\text{Shap}}$ is a i) reciprocally fair FL mechanism that admits ii) Nash equilibria iii) with strong guarantees in performance under data-sharing scheme, iv) outperforming efficiency-focused baselines.

### Breakdown
---
i) *Reciprocal Fairness*: authors provided Theorem 4.7, which guarantees that the proposed mechanism is (fully) reciprocally fair, with sound derivations and acceptable assumptions (e.g., individually rational clients).

ii) *Nash Equilibria*: authors also provided Theorem 4.3, which guarantees the existence of Nash equilibrium under common assumptions (concave payoff and convex cost), which is supported by satisfying conditions of Kakutani’s fixed-point theorem

iii) *Performance Guarantee*: authors showed in Theorem 4.8 that the proposed mechanism achieves (weak) Pareto optimality at Nash equilibrium, and also guarantees data & accuracy gains by providing related lower bounds in Theorem 4.9

iv) *Empirical Superiority*: in Table 1, the proposed mechanism consistently outperforms existing baselines across a few of vision benchmarks with a moderate number of agents (i.e., $n=100$).

**Essential References Not Discussed:**

> (Xu et al., 2021) Gradient-Driven Rewards to Guarantee Fairness in Collaborative Machine Learning
> (Zeng et al., 2022) Incentive Mechanisms in Federated Learning and A Game-Theoretical Approach
* These to references directly used Shapley value, as the same in the proposed method, for the incentive mechanism design in federated settings, but they're neither discussed nor compared.

> (Chaudhury et al., 2022) Fairness in Federated Learning via Core-Stability
* This paper defines _core-stable fairness_, which is not identical to _reciprocal fairness_, but is closely related in its similarity of fairness concepts, e.g., the existence of similar fair equilibria in the collaborative setting.

**Experimental Designs Or Analyses:**

* The evaluation setting is a bit questionable. As stated in Appendix D.1, lines 882-883, the reported metrics are the performances of _a global model_ on a _global test dataset_. Please correct me If I understood wrong: I understood that the meaning of 'global' refers to the 'server-side holdout model or dataset'. The global evaluation scheme is valid when if we'd like to evaluate the global model's performance/generalization on unseen dataset or agents. However, the target of this paper is to evaluate the utility-fairness tradeoff _as a result_ of participating in the FL process. In this regard, I believe the local evaluation scheme should be adopted. Instead of assigning an identical local test dataset (c.f line 881), let each client has their own test set and evaluate the trained global model $\theta^t$ on these local test set.
* One of the main difference between FL and distributed learning is the existence of the statistical heterogeneity, i.e., non-IID nature between local distributions. While many non-IID simulation methods exist, it seems that current experimental design does not reflect this heterogeneity simulations (McMahan et al., 2018; Hsu et al., 2019). Without this, I think we can hardly believe that the empirical validation can really be extended to the _federated learning_ setting in practice.

> (McMahan et al., 2018) Communication-Efficient Learning of Deep Networks from Decentralized Data
> (Hsu et al., 2019) Measuring the Effects of Non-Identical Data Distribution for Federated Visual Classification

**Methods And Evaluation Criteria:**

- The experimental results assume identical payoff functions learned from local training, which may not hold in practical FL settings, especially under statistical heterogeneity (i.e., non-IIDness). The variations of real-world data quality are not fully explored in this regard.
- While the simulated federated setting contains an acceptable number of clients with a proper client sampling ratio, the benchmark datasets are limited to vision classification tasks.
- Existing baseline methods seems lacking; there already are incentive mechanisms similarly designed for FL (even using Shapley value), but they are not directly compared, except (Murhekar et al., 2023)

**Other Comments Or Suggestions:**

* Please consider narrowing down the scope of the paper into _cross-silo FL_ setting (as in Zeng et al., 2022), where only a moderate number of reliable clients participate in collaboration. It's due to the statefulness design of the proposed method, such as i) the needs of full client participation as in lines 955-956 (i.e., line 7 of Algorithm 1) and ii) the existence of memory of contribution vector $\boldsymbol{s}^t$ - which typically does not hold in cross-device setting as we cannot assume repetitive participation of clients in this setup (please see Table 1 of Kairouz et al., 2019)
* Please change the global evaluation setting into local evaluation setting, and report corresponding results. In my humble opinion, it's more fair to evaluate how each client is benefited from participating in FL. If my thought is not aligned with the authors' intention, please generously enlighten me.
* Please repeat experiments with different random seeds and provide evaluation results with standard deviation.
* Please consider conducting more experiments with smaller numbers of clients, e.g. $n=2, 10, 50$. For example, with $n=2$ and a simple quadratic model on a synthetic dataset (e.g., linear/logistic regression task), it would be more visible to understand the edge of theoretical claims provided. Regarding this, please refer to the experimental setup of (Chaudhury et al., 2022)
* Please add more baselines, especially the ones in _Essential References Not Discussed_ section.
* In eq. (5), the numerator should be $\min _{\boldsymbol{s} \in \mathrm{NE}(\mathcal{M})} \Vert \boldsymbol{s} \Vert_1$.

> (Zeng et al., 2022) Incentive Mechanisms in Federated Learning and A Game-Theoretical Approach
> (Kairouz et al., 2019) Advances and Open Problems in Federated Learning
> (Chaudhury et al., 2022) Fairness in Federated Learning via Core-Stability

**Other Strengths And Weaknesses:**

Strength
- The proposed method is well-supported by structured theories and designs.
- The targeted problem is undoubtedly important for the overall welfare of the distributed learning system.

Weakness
- While theoretically sounding, the proposed method is not appealing in practical view. It requires full synchronization for acquiring $T_i, i\in[n]$, which limits its applicability in practice (e.g., cross-device FL setting, where massive number of clients exist in the system). In addition, it also requires another communication for $T_i^\epsilon$, as well as doubled local computations to obtain $T_i$ and $T_i^\epsilon$, for the calculation of the difference of Shapley shares.
- It is heartbreaking that the empirical validation of the proposed method seems lacking, questioning the statement of 'outperformance' of the proposed method compared to baselines.
- While it's inevitable to use approximations (e.g., Monte-Carlo) to calculate the Shapley value in a reasonable time, using only 5 permutations is concerning as it can introduce an approximation error. Any ablation studies on this or theoretical justification would be appreciated.

**Questions For Authors:**

* In the pseudocode of Algorithm 1, what is the meaning of 'the central server runs FedAvg'? Is it equivalent to simple averaging of local updates, i.e., $\theta^t \leftarrow \sum_{i \in R^t} T_i / k$? If so, please discard the phrase, and if not, please enlighten me.
* Can we say that $\mathcal{M}^0$ is equivalent to the `FedAvg`, as well as utilitarian setting?

**Relation To Broader Scientific Literature:**

The proposed method can contribute to achieving the global welfare of federated or collaborative machine learning systems, where many participants (i.e. agents) are willing to contribute their data for local model updates, but they are also potentially free-riders. Thus, prevention of such malfare is important in practice, and it is guaranteed in terms of _reciprocal fairness_ in the proposed method.

**Theoretical Claims:**

The theoretical expositions are well structured. I appreciate the authors' efforts since all claims are easy to follow, especially thanks to the accompanying 'implications' subsections.

---

> ### Author Rebuttal · Authors · 2025-04-01
>
> Thank you for your detailed feedback on our experiments and for appreciating our theoretical contributions.
>
> We conducted several new experiments based on your suggestions. We ran each method thrice with different seeds, and observed that our mechanism consistently outperformed baselines in data gain, accuracy, and reciprocity. All results are in: **[Experiments](https://www.dropbox.com/scl/fi/7q6m19rkdml94zo3zwrl8/submission8749_rebuttal.pdf?rlkey=797frtrobzqx62qdblefp08oy&e=1&st=9qc3dzov&dl=0)**.  We address the main weaknesses/suggestions below.
>
> **Q:** Experiments ignore statistical heterogeneity/ local evaluation
>
> **A:** We perform new experiments with non-IID agents following Ghosh et al. [1]. We have 30 clients partitioned into 3 groups of 10 each. We equally partition the images in our benchmarks into three groups and rotate the images by 10, 90, and 180 degrees respectively, giving datasets $A_1, A_2, A_3$. We then map dataset $A_j$ to group $j$, and split it into local training and test sets. Thus, this experiment accounts for non-IIDness due to the rotated images and uses local evaluation. We observe improved data gain (3x), accuracy, and reciprocity (2x) even in this non-IID setting with local evaluation. Finally, we note that identical payoffs and IID test-data is assumed in prior work [2-4].
>
> **Q:** Non-vision experiments
>
> **A:** As suggested, we implement a quadratic model on a synthetic 2-classification dataset, where inputs $X\in \mathbb{R}^{10}$ has ten features and $y\in \{0, 1\}$. We randomly generate matrix $W$, vector $b$ and number $c$ by $W_{i,j}, b_i, c\sim \mathcal{N}(0, 1)$. We have 2 agents. For non-IID distributions, we sample 1000 points uniformly for each such that 80% and 20% of agent 1's and agent 2's data have positive labels. We observe significant improvements (~30x data gain).
>
> We also request you to kindly read our response to Reviewer ir71 for experiments on a real-world healthcare dataset.
>
> **Q:** Theoretical justification for approximating Shapley value
>
> **A:** We approximate SV using standard Monte Carlo estimation [5-7]. For $n$ agents,  $m = \frac{2n}{\varepsilon^2} \ln\frac{2n}{\delta}$ samples ensure an error of $\varepsilon$ and confidence $1-\delta$. In new experiments, we set $\varepsilon = \delta = 0.1$. We will include this analysis in our paper.
>
> **Q:** Additional references
>
> **A:** Thank you for pointing out the suggested references on Shapley value in FL. We will cite them. However, our work differs in both problem setting and solution. For example, Xu et al. (2021) measure contributions via Shapley value of cosine similarities in gradient updates, while Chaudhury et al. (2022) ensure coalition stability assuming full data contribution. In contrast, we study strategic data sharing with monetary rewards. Due to the differences, we did not use them as baselines. Zeng et al. (2022) survey FL incentives and mention Wang et al. (2019), whose FL Shapley share metric aligns with our Appendix D.2 setup, and we will cite this work.
>
> **Q:** Consider narrowing the scope to cross-silo FL.
>
> **A:** Our theoretical results are general, but we agree on focusing on the cross-silo setting (Zeng et al., 2022) with a moderate number of agents.
>
> **Q:** "the proposed method... requires full synchronization for acquiring $T_i$, $i\in[n]$... it also requires another communication for $T_i^\varepsilon$, as well as doubled local computations to obtain $T_i$ and $T_i^\varepsilon$..."
>
> **A:** We agree that synchronization is a challenge in our method, but this is true in almost all cross-device FL protocols with many clients. Based on your suggestion, we will focus on cross-silo FL. We note that our protocols account for scalability by sampling only $k$ agents per round for sample updates. Since we compute $T^i_\varepsilon$ only for these $k$ clients instead of all $n$, the total computations are reduced to $n+k$ from $2n$.
>
> **Q1:** Yes, they are equivalent, and we will clarify this.
>
> **Q2:** FedBR computes the NE of the zero-payment mechanism $\mathcal{M}^0$ is the zero-payment mechanism. Each round of FedBR employs FedAvg to train the model, so they are "equivalent" at equilibrium but not identical.
>
> **References**
>
> [1] An Efficient Framework for Clustered Federated Learning, Ghosh et al. 2020
>
> [2] Mechanisms that Incentivize Data Sharing in Federated Learning, Karimireddy et al. 2022
>
> [3] Incentives in Federated Learning: Equilibria, Dynamics, and Mechanisms for Welfare Maximization, Murhekar et al. 2023
>
> [4] Incentivizing Honesty among Competitors in Collaborative Learning and Optimization, Dorner et al. 2023
>
> [5] Addressing The Computational Issues of the Shapley Value With Applications in The Smart Grid, Sasan Maleki. 2015
>
> [6] Towards Efficient Data Valuation Based on the Shapley Value, Jia et al. 2019
>
> [7] Efficient Sampling Approaches to Shapley Value Approximation, Zhang et al. 2023
>
> [8] A Principled Approach to Data Valuation for Federated Learning, Wang et al. 2020

---

> > ### Comment · Reviewer_Dvs1 · 2025-04-07
> >
> > I sincerely appreciate for the efforts and detailed responses from the authors.
> >
> > ###  Why Not Lower Score
> > ---
> > As all my concerns are clearly addressed - especially the empirical concerns - I'd like to champion this paper. Both the theoretical and the empirical justifications now seem sound to me, and I've raised the score to **accept**.
> > Based on the rebuttals, please consider adding or supplementing the following in the revised manuscript:
> > - Please adjust the scope (and accordingly polish the abstract and possibly the title) to a specific FL setup, i.e., cross-silo FL setting, to emphasize the applicability of the proposed method.
> > - Please clearly state the experimental setup, so that the proposed method seems to be clearly justified by experiments.
> > - Please also add non-vision experiments to emphasize the feasibility of the proposed methods in various practical (cross-silo) FL scenarios.
> > - Please state the limitation of the synchronization cost, which is nevertheless mitigated by authors' contribution through the subsampling scheme. Please also mention that this is acceptable in the cross-silo FL scenario since the total number of clients (i.e., $n$) is typically moderate.
> > I hope that these will help improve the presentation and communication of the research results.
> >
> > ###  Why Not Higher Score
> > ---
> > I think there still is room for improvement in following perspectives.
> > - For the reproducibility, the pseudocode of the proposed method (i.e., Algorithm 1 on page 18) could've been improved. For example, instead of providing "The central server computes (formula) as described in Section 5.1, ...", "Agent i computes (formula)", the authors can assign equation numbers to related formulae and state as: "The central server computes eq. (6)", "Agent i computes eq. (7)", for instance. Likewise, if the presentation is improved, succinct pseudocode can be moved to the main text, emphasizing easy implementation of the proposed method along with its theoretical justification.
> > - The empirical results (including those in the rebuttal) seem to have no standard deviations(or standard errors), although the authors stated that they used a new empirical setup with local evaluation. This is questionable if the authors only conduct a single run for each experiment: the number of clients is moderate in all experimental setups, thus there is little computational burden.

---

> > > ### Author Response · Authors · 2025-04-09
> > >
> > > We thank the reviewer for increasing the score and for championing the paper! Also, thank you very much for the additional suggestions, especially the one on improving the pseudocode. We will incorporate all of them in our final version of the paper.
> > >
> > > In our earlier response, we had reported the mean and median of results obtained by running each experiment three times. We also include the standard deviations in this updated [document](https://www.dropbox.com/scl/fi/ta78qamz9agmh5moo0mgg/submission8749_rebuttal_updated.pdf?rlkey=7ik5xq9ckkr9yvxl64ut9v9u3&st=zc8xq8lt&dl=0.
> > > ). We will include experimental results averaged over a larger number of runs in the final version of the paper.

---

### Decision · Program_Chairs · 2025-05-01

**Decision:**

Accept (poster)

**Comment:**

Most Reviewers voted to accept this paper based on (1) the method is well-supported by structured theories and designs, (2) the problem is important, (3) the experiements are sound, etc., which I mostly agree with. The Reviewers also raised concerns which have been properly addressed. Thus, I recommend to accept this paper.